# Attitudes towards free-roaming dogs and dog ownership practices in Bulgaria, Italy, and Ukraine

**Lauren Margaret Smith**[1]*, **Rupert Quinnell**[1], **Alexandru Munteanu**[2], **Sabine Hartmann**[2], **Paolo Dalla Villa**[3,4], **Lisa Collins**[1]*

**1** School of Biology, Faculty of Biological Sciences, University of Leeds, Leeds, United Kingdom, **2** VIER PFOTEN International, Vienna, Austria, **3** Istituto Zooprofilattico Sperimentale dell'Abruzzo e del Molise "G. Caporale", Teramo, Italy, **4** World Organization for Animal Health, OIE Sub-Regional Representation in Brussels, Brussels, Belgium

* L.Collins@leeds.ac.uk (LC); lauren.m.smith026@gmail.com (LMS)

**Data Availability Statement:** All data, code and supporting files are available from the Open Science Framework database: https://osf.io/dkxbz/

## Abstract

Free-roaming dog population management is conducted to mitigate risks to public health, livestock losses, wildlife conservation, and dog health and welfare. This study aimed to determine attitudes towards free-roaming dogs and their management and describe dog ownership practices in three European countries. We distributed an online questionnaire comprising questions relating to dog ownership practices and attitudes towards free-roaming dogs using social media. We used logistic regression and ordinal probit models to determine associations between demographic and other factors with ownership practices and attitudes towards free-roaming dogs. This study found that most surveyed respondents wanted to see a reduction in free-roaming dog numbers, and felt that this should be achieved through sheltering, catch-neuter-release, and by controlling owned dog breeding. We identified significant associations between both attitudes and ownership practices with gender, religious beliefs, age, education level, reason for dog ownership, previous experience with free-roaming dogs, and country of residence. Respondents who identified as: (i) male, (ii) holding religious beliefs, (iii) owning dogs for practical reasons, (iv) being young, or (v) having no schooling or primary education had a lower probability of neutering and a higher probability of allowing dogs to roam. Respondents who identified as: (i) female, (ii) feeling threatened by free-roaming dogs, (iii) older, or (iv) having more education had a higher probability of answering that increases in free-roaming dog numbers should be prevented. These findings can help to inform future dog population management interventions in these countries. We emphasise the importance of considering local attitudes and dog ownership practices in the development of effective dog population management approaches.

## Introduction

Domestic dogs (*Canis familiaris*) are one of the most abundant species of canids in the world, with total population estimates between 700 million to 1 billion [1, 2]. Around 75% of this

?view_only=
5b421d2488144d3a8f6a313ebe30864d.

**Funding:** LMC has received a research grant from VIER PFOTEN International (https://www.four-paws.org); and LMS's research has been funded by VIER PFOTEN International. SH and AMM are employed by VIER PFOTEN International and contributed to the conceptualisation of the study and reviewing and editing of drafts.

**Competing interests:** The authors declare that: A. M.M. and S.H. are employed by VIER PFOTEN International, a global animal welfare organisation; L.M.C has received a research grant from VIER PFOTEN International; and L.M.S.'s research has been funded by VIER PFOTEN International. This does not alter our adherence to PLOS ONE policies on sharing data and materials.

population are classified as "free-roaming", indicating their ability to roam and reproduce freely [1]. Where free-roaming dogs exist in high densities, there are important implications for public health [3–5], livestock losses [6–10] and wildlife conservation [11–13], in addition to issues for the welfare of the dogs themselves [14–17]. The management methods applied to control population sizes involve culling, reproductive control (e.g. through catch-neuter-release; CNR), and the use of shelters to house unowned or unwanted dogs [18].

Human behaviour can shape the success of a population management programme. This includes actions of local communities, the teams involved in dog population management and the governments imposing management interventions. Indeed, the World Organisation for Animal Health (OIE) has identified that understanding public attitudes is important for developing effective dog population control [19]. In order for interventions to be successful, there must be public support for both the management method and its aims (e.g. reducing or stabilising the number of free-roaming dogs). Different communities may have different attitudes towards free-roaming dogs and management methods due to culture, religion, and the specific risks to humans, wildlife, livestock, and other companion animals in the area. Organisations involved in dog population management should consider these cultural, religious, and risk factors to ensure interventions are effective. For example, free-roaming dog populations can be an important part of a community, providing protection to people and livestock [17]. Where management methods aim to reduce free-roaming dog numbers, there may still be demand for dogs in a community. Unless this demand reduces, new dogs may be acquired to replace those removed by population management (either bought, adopted, or by uncontrolled immigration of free-roaming dogs). Prior to implementing management interventions, the level of acceptance of free-roaming dogs in the area should be gauged (i.e. determine whether the public prefer to have fewer free-roaming dogs in the community) so that those involved in dog population management can work towards a goal that benefits the community.

Dog ownership practices can also influence the success of population management. Encouraging responsible ownership practices is included as an objective in Chapter 7.7 of the OIE Terrestrial Animal Health Code on free-roaming dog population control [19]. The OIE describes the requirements of responsible ownership as: "*When a person takes on the ownership of a dog, there should be an immediate acceptance of responsibility for the dog, and for any offspring it may produce, for the duration of its life or until a subsequent owner is found*" [19]. Unrestricted owned dogs (free-roaming owned) and abandoned dogs are sources of free-roaming dog population increase [19]. Dog ownership practices that allow owned dogs to roam and do not prevent reproduction can hinder efforts to control free-roaming dog populations. Those involved in dog population management must determine the extent to which owned dogs contribute to the free-roaming dog population so that management interventions can be tailored appropriately (e.g. by enforcing responsible ownership through legislation and education programmes).

Questionnaire surveys are frequently used to gain insight into attitudes, opinions, behaviours, and the demographic and sociological factors associated with these. In terms of dog population management, different attitudes, opinions and behaviours about and towards dogs have been associated with responder gender [20–24], age [20, 22, 25], education [21, 22, 26], and previous life experiences (e.g. experience of keeping dogs in childhood) [27]. Questionnaires aiming to describe dog ownership, attitudes, and knowledge have been conducted in many countries around the world, but few published studies have been carried out in European countries.

This study determines attitudes towards the presence of free-roaming dogs and of dog ownership practices in three European countries–Bulgaria, Italy, and Ukraine. We selected these focal countries due to the networks established with collaborating organisations (VIER

PFOTEN International and Istituto Zooprofilattico Sperimentale dell'Abruzzo e del Molise "Giuseppe Caporale"; IZSAM) that provided local knowledge to facilitate data collection. The focal countries are culturally and environmentally distinct, allowing comparison of the collected data between different countries within Europe.

In this study, we define attitudes as the thoughts, feelings, and opinions of respondents, as reported in the questionnaire. We define ownership practices as the actions taken to acquire, provide care, and relinquish ownership of dogs, as reported in the questionnaire. The objectives of this study are to: (i) determine attitudes towards the presence of free-roaming dogs; (ii) determine local ownership practices and attitudes, including whether owned dogs were free-roaming or neutered, the level of dog abandonment, and the reasons for dog abandonment; and (iii) investigate whether demographic and other factors (including age, gender, education level, religious beliefs, and previous experience with dogs) influence ownership practices and attitudes towards free-roaming dogs. This information can inform interventions so that education campaigns can target groups who are at-risk of irresponsible dog ownership behaviours [20, 26], as well as provide a baseline for evaluating the impact of interventions on human behaviour and attitudes [26].

## Materials and methods

### Study design

This was a cross-sectional study, with target populations of Bulgaria, Italy, and Ukraine. The study populations were residents who used social media. Participants were recruited through social media using an online questionnaire (Online Surveys [28]) that was open between the 8th of March 2019 and the 21st of December 2019, available in four languages: Bulgarian, Italian, Ukrainian, and Russian. The social media outlets used to distribute the questionnaire included Facebook [29] and Twitter [30]. Facebook advertising was used to increase the visibility of the questionnaire to the study population and increase the number of respondents. Facebook advertising targeted Facebook users who: (i) were recorded in their online profile as living in Bulgaria, Italy, or Ukraine; and (ii) were over the age of 18. The Facebook adverts invited respondents to provide their opinion on free-roaming dogs and dog ownership practices (see S1 File for English translation of adverts). All adverts were used in all countries to an equal extent, though it is not possible to know which advert respondents had seen. Minimum sample sizes were calculated for the three study areas, using Eq 1. A sample size of 385 respondents per study country was necessary to provide estimates with a 95% confidence interval that was within a 5% margin of error.

Eq 1. Sample size calculation

$$Sample\ size = \frac{\frac{z^2 \times p(1-p)}{e^2}}{1 + \left(\frac{z^2 \times p(1-p)}{e^2 N}\right)}$$

Where N = population size, e = margin of error, z = z-score, p = population proportion.

### Ethical approval

Prior to completing the questionnaire, all participants gave consent within the online survey through a digital signature by selecting yes in a tick box to confirm that they (i) had read and understood the information sheet explaining the project (S2 File), (ii) understood that they had the opportunity to ask questions about the project, (iii) knew they could withdraw from completing the questionnaire, prior to submitting, (iv) agreed for their responses to be collected, stored, and analysed in an anonymised form for the purpose of reports and publication

(see S2 File for information). Those who did not consent were not able to complete the questionnaire and were therefore not included in the study. No directly identifiable information was collected; all data obtained remains anonymous. Participants were able to withdraw from the questionnaire prior to completion, but as the data was collected anonymously, participants could not withdraw after the questionnaire was submitted. No minors were included in this study, as respondents who reported being under the age of 18 were not able to complete the online questionnaire. The study was approved by the University of Leeds Ethical Committee (reference BIOSCI 17–003).

## Questionnaire design

The questionnaire was developed in English and translated into Bulgarian, Italian, Ukrainian, and Russian. The questionnaire comprised closed questions regarding the respondents' attitudes towards free-roaming dogs and their management. Respondents that reported owning a dog were asked to also complete questions relating to dog ownership practices. Likert-type scales were used to estimate the level of agreement with specific questions. The questionnaire consisted of three sections: (i) socio-demographic information of the respondent (all respondents); (ii) ownership practices (only dog owners); and (iii) attitudes towards the presence of free-roaming dogs and the management of the free-roaming dog population (all respondents). A copy of the questionnaire in English can be found in S2 File.

## Statistical analyses

All predictor and response variables are described in Table 1. Bernoulli logistic regression models were used to test associations between demographic parameters and respondent

**Table 1. Response and predictor variables (self-reported responses to questions) included in the statistical analyses and their levels.**

| Variables | Levels |
|---|---|
| *Age* | * *18–24, 25–34, 35–44, 45–54, 55–64, 65–74 and 75 and above* |
| *Children in household* | *Children in household, no children in household* |
| *Country* | *Bulgaria, Italy, Ukraine* |
| *Dog ownership* | *Dog owner, non-dog owner* |
| *Education status* | * *No education, primary, secondary, tertiary* |
| *Gender* | *Male, Female,* NA (including option *Other*) |
| *Neutering status of owned dogs* | *Neutered, not neutered* |
| *Owning a dog for practical reasons* | *Practical, not practical* |
| *Religious belief* | *Religious, non-religious* |
| *Feeling physically threatened by dogs on the street* | * *Strongly disagree, disagree, neither agree nor disagree, agree, strongly agree* |
| *Been attacked by dogs on the street* | *Been attacked, not been attacked* |
| *Respondent or family members have been bitten by dogs on the street in last 12 months* | *Been bitten, not been bitten* |
| *Roaming status of owned dogs* | *Never, Sometimes, Always* |
| *I do not like the presence of free-roaming dogs around my home or work* | *Strongly disagree, disagree, neither agree nor disagree, agree, strongly agree* |
| *Should an increase in dogs on the street be prevented?* | *Yes, No* |
| *Would you prefer to see dogs on the street?* | *No free-roaming dogs, fewer free-roaming dogs, do not mind free-roaming dogs, more free-roaming dogs* |

* Ordinal predictor variables analysed as continuous variables in statistical models.

experience on the binary response variables: (i) *Neutering status of owned dogs* (binary response: neutered/not neutered); and (ii) respondents' answers to the question *"Do you think an increase in dogs on the street should be prevented?"* (binary response: Yes/no). Ordinal probit models [31] were used to test associations between demographic parameters and respondent experience on: (i) *Roaming status of owned dogs*; (ii) *I do not like the presence of free-roaming dogs around my home or work*; and (iii) respondents' answers to the question "*Would you prefer to see: no free-roaming dogs, fewer free-roaming dogs, do not mind free-roaming dogs, more free-roaming dogs*". Ordinal variables are categorical variables with a natural order, for example Likert-type scales [32]. Ordinal variables are assumed to have an underlying continuous latent variable that cannot be measured directly (e.g. the attitude of a respondent). This underlying latent variable is therefore split into discrete options that can be measured (e.g. *Strongly agree* or *Agree*). The intervals between these discrete options may not be equal (i.e. not equidistant), an assumption required by metric models [33], and responses to ordinal questions may have non-normal distributions. Ordinal predictor variables can be problematic if analysed metrically, leading to Type I (false positive) and Type II (false negative) errors [31]. Ordinal models deal with issues in potential non-equidistant responses and non-normal distributions.

**Dog ownership practices.** Model 1 tested associations between demographic parameters and respondent experience with neutering of owned dogs using a Bayesian Bernoulli logistic regression model. The response variable was *neutering status of owned dogs* with fixed effects of *gender, age, education status, religious belief, owning a dog for practical reasons* and *country* (Table 1). Model 2 tested associations between demographic parameters and respondent experience with the roaming status of owned dogs using a Bayesian ordinal probit model. The response variable was *roaming status of owned dogs* and fixed effects were the same as for Model 1.

**Attitudes towards free-roaming dogs.** Model 3 tested associations between demographic parameters and respondent experience with agreement to the statement *I do not like the presence of free-roaming dogs around my home or work* using a Bayesian ordinal probit model. The response variable was *I do not like the presence of free-roaming dogs around my home or work* and fixed effects were *dog ownership, gender, age, education status, children in household, feeling physically threatened by dogs on the street, been attacked by dogs on street, respondent or family members have been bitten by dogs on the street in last 12 months*, and *country*.

Model 4 tested associations between demographic parameters and respondent experience with the question *Do you think an increase in dogs on the street should be prevented?* using a Bayesian Bernoulli logistic regression model. The response variable was *should an increase in dogs on the street be prevented*, with fixed effects the same as in Model 3.

Model 5 tested associations between demographic parameters and respondent experience with response to the question *Would you prefer to see: no free-roaming dogs, fewer free-roaming dogs, do not mind free-roaming dogs, more free-roaming dogs* using a Bayesian ordinal probit model. The response variable was *Would you prefer to see dogs on the street*, with fixed effects the same as in Model 3.

To fit the statistical models using a Bayesian analysis framework, the package "**brms**" version 2.12.0 [34] was used in R version 3.6.1 [35]. All models were run with four chains, each with 2000 iterations (1000 used for warmup and 1000 for sampling). Thinning was set to one. The total number of post-warmup samples was 4000. Where a response was missing (i.e. a respondent did not answer a question), the response was omitted from the statistical analysis (see S1 Table for number of no responses per variable).

Collinearity in the predictor variables was checked using the "vif" function in R package "car" [36] and values lower than three were considered not collinear. Model parameters were summarised by the mean and 95% credible intervals of the posterior distribution (CI; 95%

most probable values). A significant association was determined if the 95% credible intervals of the posterior distribution did not contain zero on the log odds or probit scale. Probabilities were converted from the logit scale to the probability scale by $\exp(x)/_1+(\exp(x))$, and are converted to odds using $\exp(x)$, where $x$ is the posterior value on the logit scale.

## Results

### Descriptive analyses

**Demographics.** The numbers of respondents were 5,434 in Bulgaria, 3,468 in Italy, and 19,323 in Ukraine. All demographic information is provided in S2 Table. Respondents were from multiple regions within Bulgaria, Italy, and Ukraine (see S3 to S5 Tables). A broad range of ages between 18 and 64 were represented in all three study countries. Most respondents were female in all three study countries (87.5%, n = 4,754 of 5,434, in Bulgaria, 83.1%, n = 2,882 of 3,468 in Italy, and 87.1%, n = 16,832 of 19,323 in Ukraine). In Bulgaria 68.9% (n = 3,743 of 5,434), in Italy 42.0% (n = 1,457 of 3,468), and in Ukraine 67.3% (n = 13,011 of 19,323) of the respondents considered themselves to be religious. In Bulgaria 36% (n = 1,970 of 5,434), Italy 43% (n = 1,474 of 3,468) and Ukraine 57% (n = 10,928 of 19,323) of respondents lived in households with children.

**Ownership practices.** Sixty-five percent of respondents in Bulgaria (n = 3,528 of 5,434), 75% in Italy (n = 2,581 of 3,468) and 56% in Ukraine (n = 10,797 of 19,323) reported owning a dog. The main reason for dog ownership in all three study countries was for pleasure and company (Bulgaria 85.5%, n = 3,017 of 3528, Italy 87.7%, n = 2,263 of 2,581, Ukraine 70.7%, n = 7,631 of 10,797; see S6 Table for detailed responses on ownership practices). In Italy, a higher percentage of respondents acquired their dog from a dog shelter (38.1%, n = 986 of 2,581), compared to in Bulgaria (9.7%, n = 341 of 3,528) and Ukraine (9.9%, n = 1,070 of 10,797) (Fig 1). In Bulgaria and Ukraine, more respondents found their dog on the street (Bulgaria 35.5%, n = 1,252 of 3,528, and Ukraine 34.6%, n = 3,734 of 10,797) or received their dog from friends/family (Bulgaria 32.6%, n = 1,149 of 3,528 and Ukraine 27.9%, n = 3,015 of 10,797). More respondents in Italy answered that they prevent their dog from breeding through neutering (65.4%, n = 1,689 of 2,581), compared to 40.4% (n = 1,424 of 3,528) in Bulgaria and 35.4% (n = 3,828 of 10,797) in Ukraine. When asked the reason why respondents did not prevent breeding, 37.6% (n = 82 of 218) of respondents in Bulgaria, 34.6% (n = 75 of 217) in Italy, and 13.7% (n = 242 of 1770) in Ukraine answered: "*A dog should reproduce at least once*" (Fig 1). When respondents were asked if they allowed their dog to roam outside unsupervised, 59.0% (n = 2,082 of 3,528) in Bulgaria, 92.1% (n = 2,377 of 2,581) in Italy and 79.4% (n = 8,571 of 10,797) in Ukraine responded *Never*, and 29.5% (n = 1,039 of 3,528) in Bulgaria, 6.3% (n = 162 of 2,581) in Italy and 16.3% (n = 1764 of 10,797) in Ukraine responded *Sometimes*.

Most respondents in all study countries responded that they had never given up a dog (Bulgaria 98.5%, n = 3,474 of 3,528, Italy 92.4%, n = 2,386 of 2,581, and Ukraine 92.2%, n = 9950 of 10,797). Those respondents who had given up a dog mostly answered that this was because of an *Animal behavioural problem* (Bulgaria 27.3%, n = 9 of 33, Italy 36.5%, n = 66 of 179, and Ukraine 23.8%, n = 161 of 676), or *Other* reason (Bulgaria 39.4%, n = 13 of 33, Italy 57.5%, n = 104 of 179, and Ukraine 45.3%, n = 306 of 676) (Fig 1), such as family illness; a change in circumstances (e.g. birth of new child in home); moving home; owners going on a long trip away; the dog having puppies; or the dog not getting along with other dogs in the household.

**Attitudes.** In Bulgaria and Ukraine, high percentages of respondents had seen a free-roaming dog on the day they filled in the questionnaire (73.3%, n = 3,983 of 5,434, and 77.3%, n = 14,934 of 19,323 respectively), compared to only 15.4% (n = 534 of 3,468) of respondents

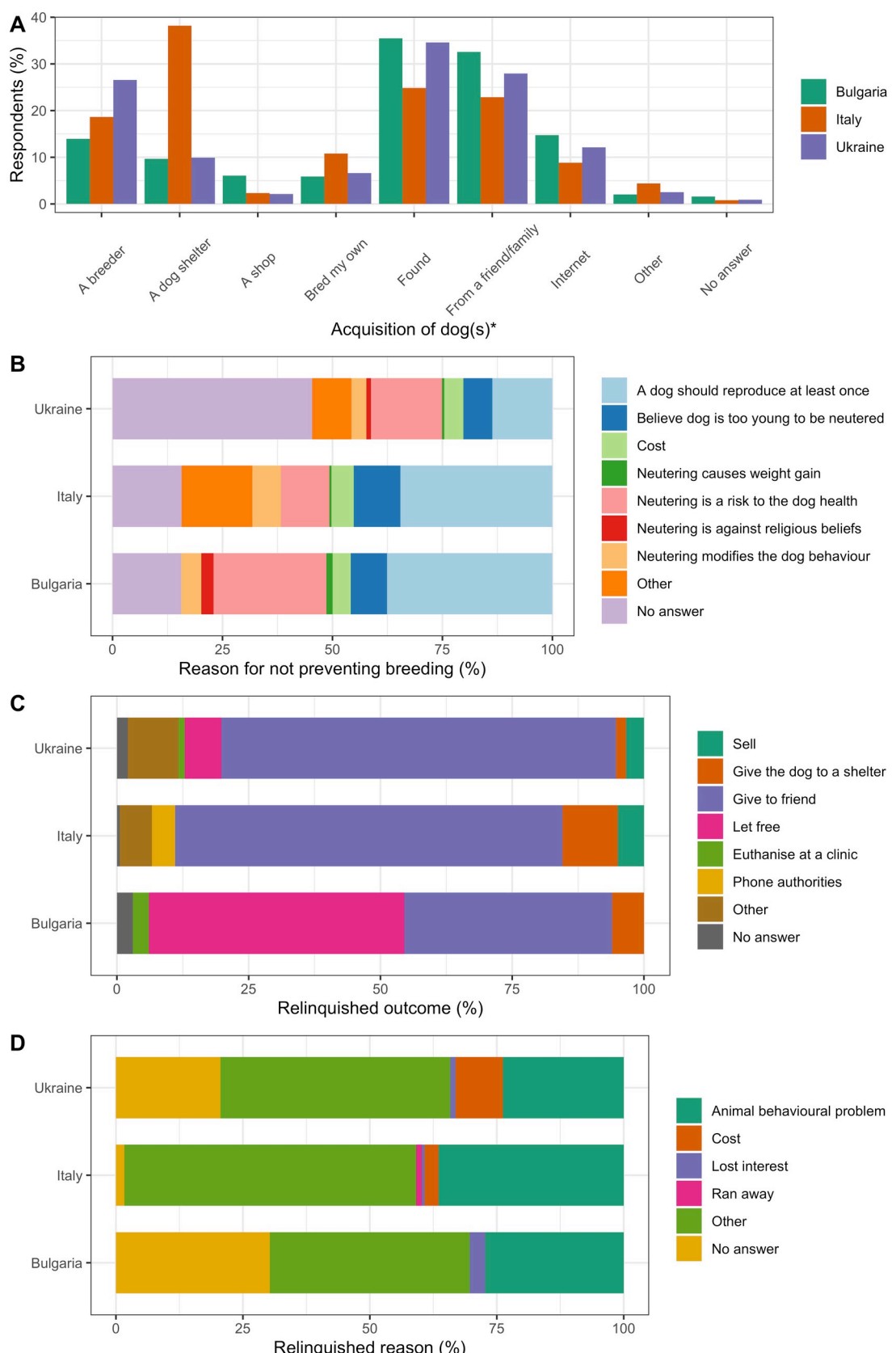

**Fig 1. Ownership practices of respondents in Bulgaria, Italy, and Ukraine.** The percentage of respondents who answered each of the answer options regarding (A) acquisition of dog, (B) reasons for not preventing breeding, (C) the outcome of the dog, and (D) reason for relinquishment. * Multi answer question: Percentage *of respondents who selected each answer option (i.e. 100% would indicate that all respondents chose this option).*

in Italy (Fig 2; see S7 Table for detailed results). A higher percentage of respondents in Bulgaria (21.6%, n = 1,174 of 5,434) and Ukraine (26.5%, n = 5,129 of 19,323) had been attacked by dogs on the street ever in their lifetime, compared to few (4.2%, n = 147 of 3,468) in Italy. Higher percentages of respondents in Bulgaria answered that they provided care to free-roaming dogs by giving food (90.6%, n = 4,911 of 5,434), water (71.0%, n = 3,847 of 5,434), and shelter (34.8%, n = 1,886 of 5,434), compared to Italy (53.7%, n = 1831 food, 44.2%, n = 1508 water, 19.0%, n = 647 of 3468 shelter) and Ukraine (67.5%, n = 13,045 food, 29.6%, n = 5,721 water and 9.7%, n = 1,882 of 19,323 shelter) (S6 Table).

When respondents were asked their level of agreement with the statement "*I do not like free-roaming dogs being present in the streets around my home or work*", responses were varied across the full range of options between strongly disagree and strongly agree in Bulgaria and Ukraine (varying between 14 and 25% for all answer options) (Fig 2). Most respondents in Italy strongly disagreed with this statement (35.8%, n = 1,242 of 3,468). In all three study countries, most respondents disagreed (Bulgaria 20.8%, n = 1,130 of 5,434, Italy 19.3%, n = 671 of 3,468, and Ukraine 25.3%, n = 4,890 of 19,323) and strongly disagreed (Bulgaria 42.2%, n = 2,295 of 5,434, Italy 56.4%, n = 1,956 of 3,468 and Ukraine 31.6%, n = 6,097 of 19,323) with the statement "*I feel physically threatened by free-roaming dogs*".

Respondents answered most often that the municipality government and volunteer organisations should be responsible for managing the free-roaming dog population (Fig 3; S6 Table). Respondents most often answered that they would like to see *no* (Bulgaria 52.4%, n = 2,848 of 5,434, Italy 70.2%, n = 2,435 of 3,468, and Ukraine 45.2%, n = 8,740 of 19,323) and *fewer* (Bulgaria 32.8%, n = 1,780 of 5,434, Italy 24.3%, n = 841 of 3,468, and Ukraine 40.6%, n = 7,846 of 19,323) free-roaming dogs. Respondents who answered that they would like to see *no* or *fewer* free-roaming dogs answered that this should be achieved through sheltering, CNR, and controlling the breeding of owned dogs (Fig 3). Few answered that the free-roaming dog population should be reduced through culling (Bulgaria 1.7%, n = 92 of 4628, Italy 1.6%, n = 56 of 3276, and Ukraine 7.3%, n = 1,216 of 16586).

## Statistical analyses

All models converged (for all parameters Rhat = 1.00 and effective sample size >1000, see Supplementary information). There was no collinearity in the predictor variables (all values less than three). All raw model results (including the posterior mean values, standard deviations and 95% credible intervals, the 2.5% and 97.5% percentiles of the posterior distribution) are presented in S8 to S12 Tables. Estimates for mean and 95% CIs for probabilities are reported for each model and presented in Table 2. Odds ratios (OR) are reported for predictor variables in the Bernoulli logistic regression models (Models 1 and 4).

**Summary of statistical associations.** Respondents were less likely to answer that they neutered their dog(s) and more likely to answer that they allow their dog(s) to roam if they identified as (i) male, (ii) religious, (iii) owning dogs for practical reasons, (iv) young, and (v) having no schooling or primary education. Respondents were more likely answer that an increase in free-roaming dogs should be prevented if they identified as (i) female, (ii) feeling threatened by free-roaming dogs, (iii) older, and (iv) having more education. Below we report the detailed statistical findings.

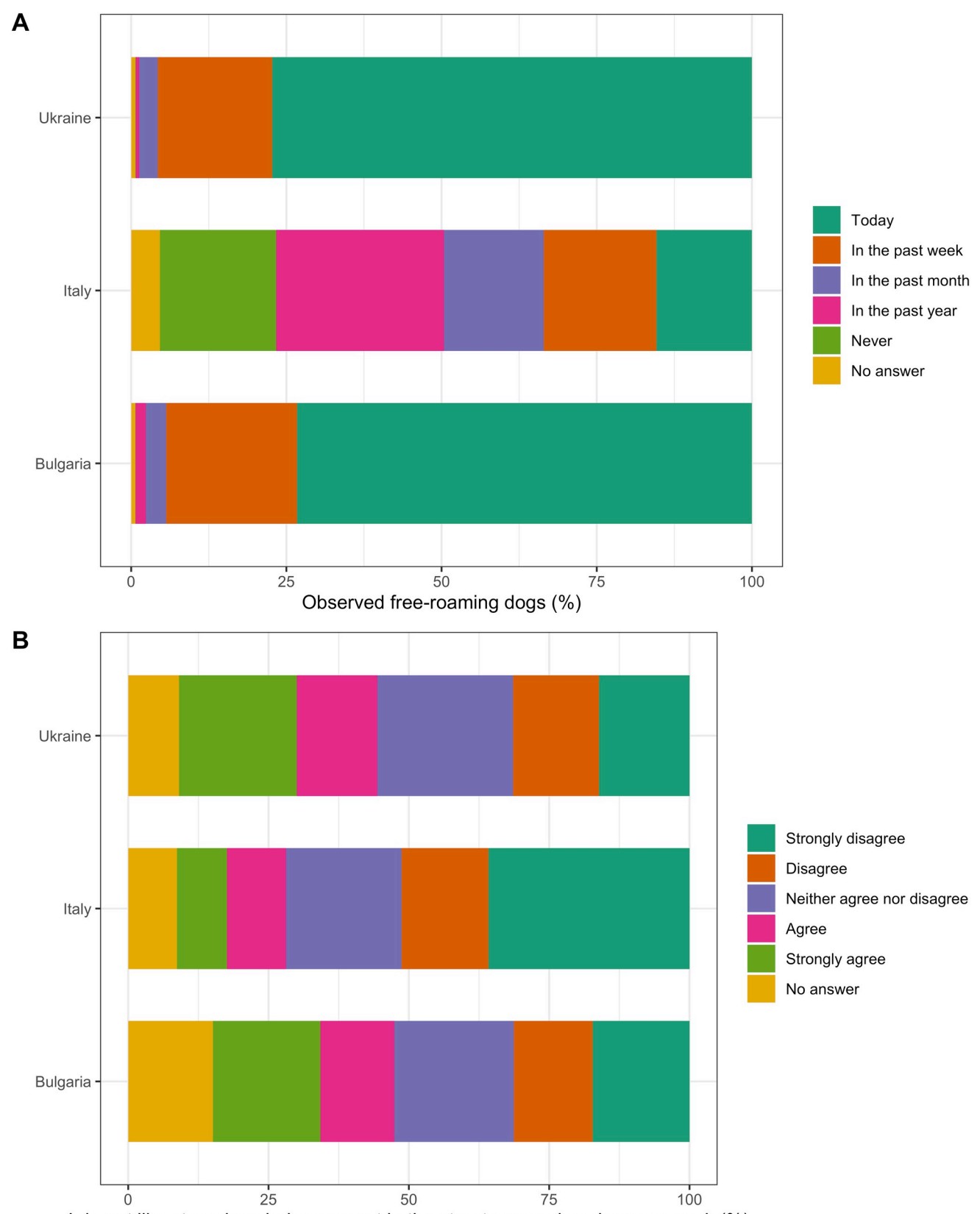

**Fig 2. Attitudes of respondents towards free-roaming dogs in Bulgaria, Italy and Ukraine.** The percentage of respondents who answered each of the answer options regarding (A) observation of free-roaming dogs and (B) agreement with statement "*I do not like free-roaming dogs being present around my home or work*".

**Respondent demographics and owned dog neutering status.**   *Gender*, *age*, e*ducation level*, *owning a dog for practical reasons*, *religious beliefs* and *country* had significant associations with the *neutering status of owned dogs* (S8 Table). Probabilities of neutering are presented in Table 2. Female respondents had a higher probability of neutering, compared to males (OR 1.47; 95% CI 1.28–1.64). Holding religious beliefs (OR 0.66, 95% CI 0.61 to 0.72) and owning dogs for practical reasons (i.e. guarding or hunting, compared to for pleasure and company; OR 0.49, 95% CI 0.43 to 0.54) were both negatively associated with neutering. Respondent age (OR 1.15; 95% CI 1.12–1.18) and education level (OR 1.29; 95% CI 1.17–1.41) were both positively associated with neutering (i.e. the older and more educated a respondent, the more likely they were to neuter). Respondents from Italy had a higher probability of neutering compared to Bulgaria (OR 2.32; 95% CI 2.05–2.62) and Ukraine (OR 2.73; 95% CI 2.44–3.01). Respondents from Ukraine had a lower probability of neutering compared to Bulgaria (OR 0.37; 95% CI 0.33–0.40).

**Respondent demographics and owned dog roaming status.**   *Gender*, a*ge*, *education level*, *owning a dog for practical reasons*, *religious beliefs*, and *country* had significant associations with the *roaming status of owned dogs* (S9 Table). Probabilities of answering *Never* allow dog to roam for predictor variables are presented in Table 2. Females had a higher probability of answering that they *Never* allowed their dog to roam. Respondents who held religious beliefs, and respondents who owned dogs for practical reasons were less likely to answer *Never*. Age of respondent was positively correlated with answering *Never* (i.e. older respondents were less likely to allow their dog to roam). Increased education level of the owner was positively associated with answering *Never* (i.e. respondents with higher levels of education were less likely to allow their dog to roam). Respondents from Italy had the highest probability of answering *Never*, respondents in Bulgaria had the lowest probability of answering *Never*.

**Respondent demographics and experience with free-roaming dogs and respondent answers to "*I do not like the presence of free-roaming dogs around home or work*".**   Predictor variables *gender*, *age*, *owning a dog for practical reasons*, *feeling threatened by dogs on the street*, *having been attacked by dogs on the street*, *respondent or family members have been bitten by dogs on the street in last 12 months*, and *country* were significantly associated with the statement *I do not like the presence of free-roaming dogs around my home or work* (S10 Table). Probabilities for answering *Strongly agree* for predictor variables are presented in Table 2. Female respondents had a lower probability of agreeing with the statement. Respondents who answered *Yes* to the question *Have you ever been attacked by dogs on the street?* had a higher probability of agreeing with the statement. Respondents who answered *Yes* to the question *Have you or your family members been bitten in the last 12 months?* had a higher probability of agreeing with the statement. Respondent age was positively associated with agreement to the statement (i.e. older respondents were more likely to agree). Agreement with the statement *I feel physically threatened by dogs on the street* was positively associated with agreement with the statement *I do not like the presence of free-roaming dogs around my home or work* (i.e. respondents who felt threatened were more likely to agree with the statement that they did not like the presence of dogs around their home or work). Respondents from Italy had the lowest probability of answering *Strongly* agree, and respondents from Bulgaria had the highest probability of answering *Strongly* agree.

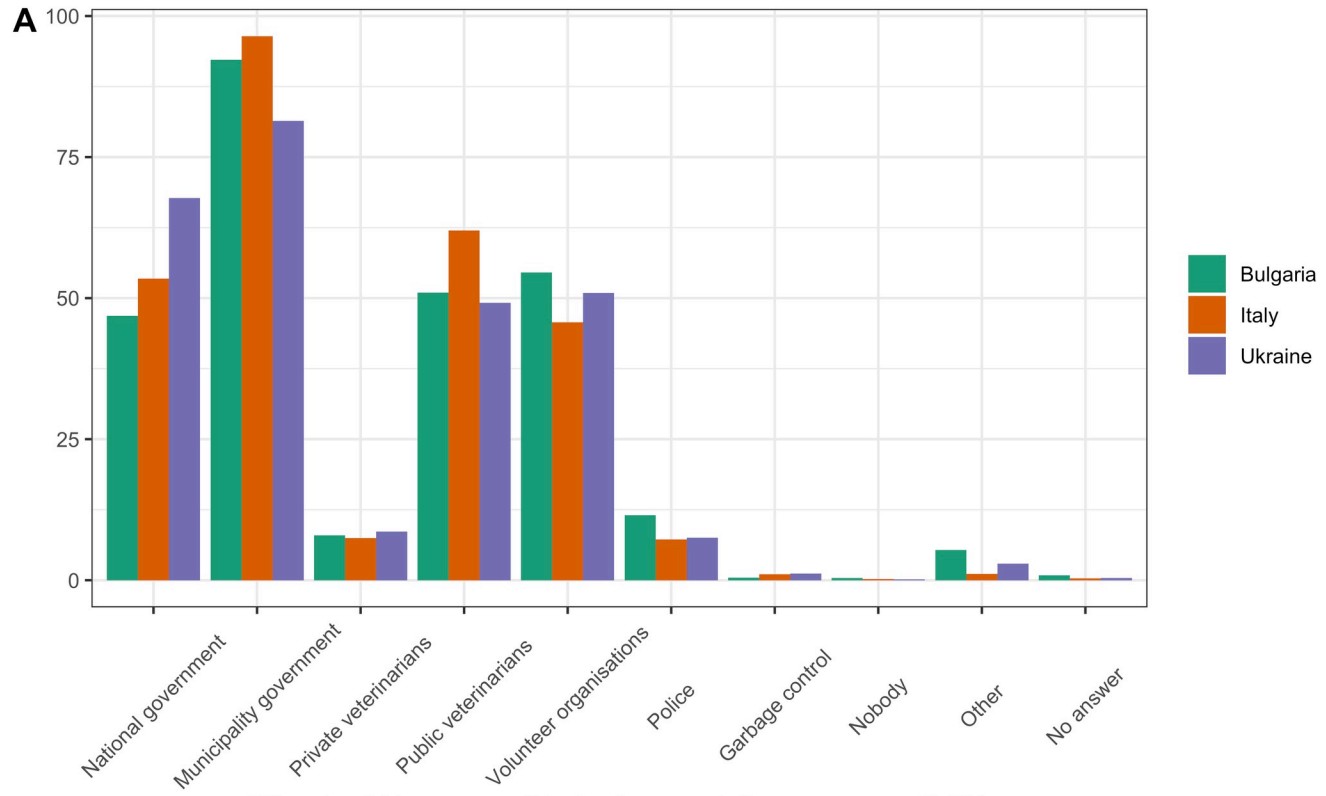

Who should be responsible for dog population managment? (%)

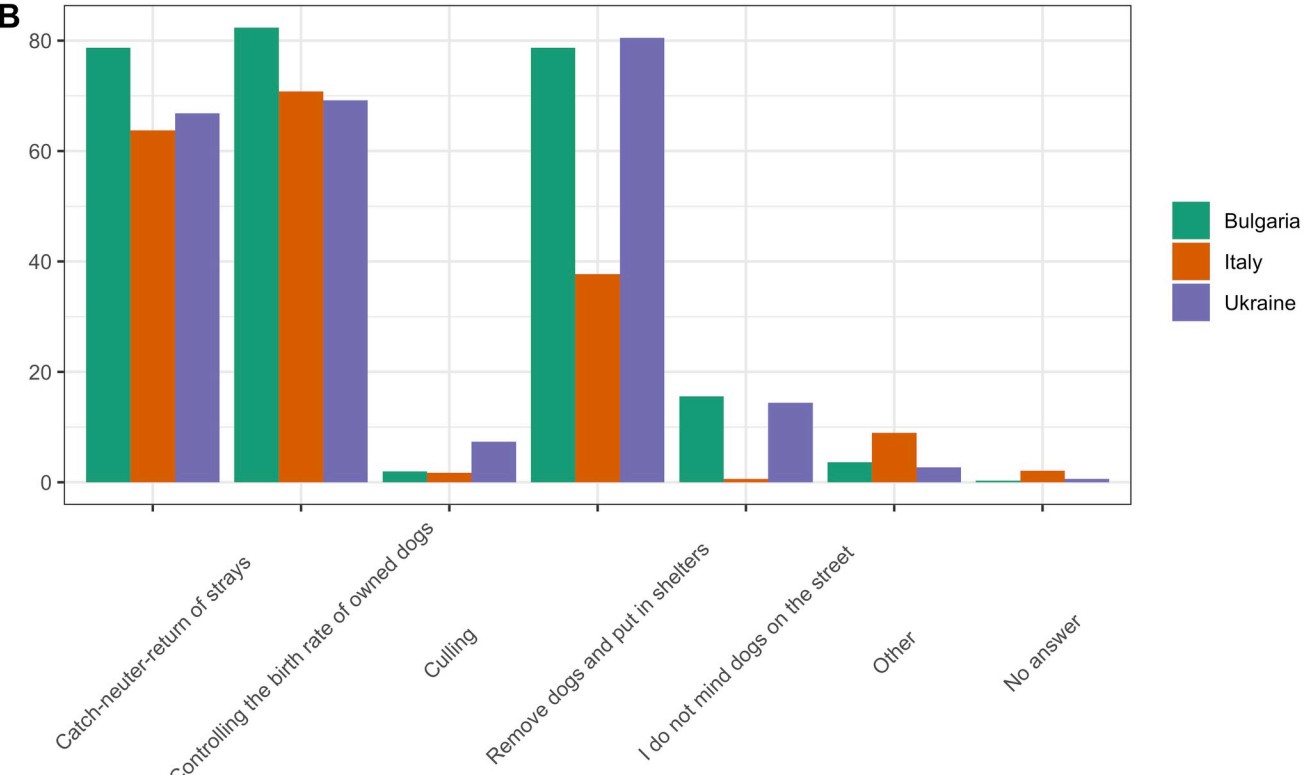

How should free-roaming dogs be reduced? (%)

**Fig 3. Attitudes of respondents towards dog population management in Bulgaria, Italy, and Ukraine.** The percentage of respondents who answered each of the answer options for: (A) who should be responsible for dog population management? and (B) how should free-roaming dogs be reduced? These were multi answer questions: Percentage of respondents who selected each answer option (i.e. 100% would indicate that all respondents chose this option).

There was no evidence of significant associations between: *dog ownership*, *education level*, and *children in household* and agreement with the statement *I do not like free-roaming dogs present around my home or work* (S10 Table).

**Respondent demographics and experience with free-roaming dogs and respondent answers to "*Should an increase in free-roaming dogs should be prevented*?".** There were significant associations between *gender*, *age*, *education level*, and *feeling threatened by dogs on the street* and answering *Yes* to the question *Do you think an increase in dogs on the street should be prevented*? (S11 Table). Female respondents had a higher probability of answering *Yes* (OR 2.14; 95% CI 1.67–2.65). There was a positive association between answering *Yes* and respondents' agreement with the statement "*I feel physically threatened by dogs on the street*" (OR 1.53; 95% CI 1.37–1.68); their age (OR 1.11; 95% CI 1.03–1.19); and education (OR 1.54; 95% CI 1.24–1.83).

There was no evidence of significant associations between *dog ownership*, *children in household*, *having been attacked by dogs on the street*, *respondent or family members have been bitten by dogs on the street in last 12 months* or *country* and answering *Yes* to the question *Do you think an increase in dogs on the street should be prevented*? (S11 Table).

**Associations between demographic parameters and respondent experience with the question "Would you prefer to see: No free-roaming dogs, fewer free-roaming dogs, do not mind free-roaming dogs, more free-roaming dogs".** *Dog ownership*, *gender*, *age*, *feeling threatened by dogs on the street*, *having been attacked by dogs on the street*, *respondent or family members have been bitten by dogs on the street in last 12 months*, and *country* had significant associations with response to this question regarding preference for observing free-roaming dogs (S12 Table). Probabilities for answering *No free-roaming dogs* for predictor variables are presented in Table 2. Male respondents had a lower probability of answering *No free-roaming dogs* (Table 2). Dog owners had a higher probability of answering *No free-roaming dogs*. Respondents who answered *Yes* to the question "*Have you ever been attacked by dogs on the street*?", or *Yes* to the question "*Have you or your family members been bitten in the last 12 months*" or had children in their household had a higher probability of answering *No free-roaming dogs*. Agreement with the statement "*I feel physically threatened by dogs on the street*" was positively correlated with answering *No free-roaming dogs* (i.e. respondents who feel threatened by dogs on the street are more likely to answer *No free-roaming dogs*). Respondent age was positively correlated with answering *No free-roaming dogs* (i.e. older respondents had a higher probability of preferring to see *No free-roaming dogs*). Respondents in Italy had the highest probability of answering *No free-roaming dogs*, and respondents in Ukraine had the lowest probability of answering *No free-roaming dogs*.

There was no evidence of significant associations with children in household and education level on the probability of preference of observing free-roaming dogs (S12 Table).

## Discussion

This study provides a summary of attitudes towards the presence and management of free-roaming dogs, and of dog ownership practices in Bulgaria, Italy, and Ukraine. We have also identified associations between responder characteristics and ownership practices or attitudes towards free-roaming dogs and their management in Bulgaria, Italy, and Ukraine. Respondents who identified as male, religious, owning dogs for practical reasons, young, or

**Table 2. Associations between predictor and outcome variables on the probability scale.**

| Predictor variable | | Model 1 | Model 2 | Model 3 | Model 4 | Model 5 |
|---|---|---|---|---|---|---|
| | | Probability of neutering (95% CI) | Probability of answering *Never* allow dog to roam (95% CI) | Probability of answering *Strongly agree* to question "*I do not like the presence of free-roaming dogs around home or work*" (95% CI) | Probability of answering *Yes* to the question "*Should an increase in free-roaming dogs be prevented*" (95% CI) | Probability of answering *No free-roaming dogs* to question "*Would you prefer to see*"(95% CI) |
| Gender | Male | 0.38 (0.35–0.41) | 0.74 (0.71–0.76) | 0.17 (0.16–0.18) | 0.97 (0.96–0.97) | 0.53 (0.51–0.55) |
| | Female | 0.48 (0.47–0.49) | 0.81 (0.80–0.82) | 0.15 (0.14–0.15) | 0.98 (0.98–0.99) | 0.60 (0.59–0.61) |
| Religious belief | Religious | 0.44 (0.43–0.45) | 0.79 (0.78–0.80) | NI | NI | NI |
| | Non-religious | 0.54 (0.52–0.56) | 0.83 (0.82–0.85) | NI | NI | NI |
| Reason for dog ownership | Practical | 0.31 (0.28–0.33) | 0.71 (0.69–0.74) | NI | NI | NI |
| | Non-practical | 0.48 (0.47–0.49) | 0.81 (0.80–0.82) | NI | NI | NI |
| Been attacked by dogs on the street | Attacked | NI | NI | 0.16 (0.15–0.17) | 0.98 (0.98–0.99) | 0.61 (0.59–0.62) |
| | Not attacked | NI | NI | 0.15 (0.14–0.15) | 0.98 (0.98–0.99) | 0.59 (0.58–0.59) |
| Respondent or family members have been bitten by dogs on the street in last 12 months | Bitten | NI | NI | 0.19 (0.18–0.21) | 0.98 (0.98–0.99) | 0.65 (0.63–0.67) |
| | Not bitten | NI | NI | 0.15 (0.14–0.15) | 0.98 (0.98–0.99) | 0.58 (0.57–0.59) |
| Dog ownership | Yes | NI | NI | 0.15 (0.15–0.16) | 0.98 (0.98–0.99) | 0.59 (0.59–0.60) |
| | No | NI | NI | 0.15 (0.14–0.16) | 0.98 0.98–0.99) | 0.56 (0.55–0.57) |
| Children in household | Yes | NI | NI | 0.15 (0.15–0.16) | 0.98 (0.98–0.99) | 0.59 (0.58–0.60) |
| | No | NI | NI | 0.15 (0.14–0.15) | 0.98 (0.98–0.99) | 0.59 (0.57–0.59) |
| Age | 18–24 | 0.40 (0.39–0.42) | 0.78 (0.76–0.79) | 0.14 (0.14–0.15) | 0.98 (0.98–0.98) | 0.57 (0.56–0.58) |
| | 25–34 | 0.44 (0.43–0.45) | 0.79 (0.78–0.80) | 0.15 (0.14–0.15) | 0.98 (0.98–0.99) | 0.58 (0.57–0.59) |
| | 35–44 | 0.47 (0.46–0.48) | 0.80 (0.80–0.81) | 0.15 (0.15–0.16) | 0.98 (0.98–0.99) | 0.59 (0.58–0.60) |
| | 45–54 | 0.51 (0.49–0.52) | 0.82 (0.81–0.83) | 0.16 (0.15–0.16) | 0.99 (0.98–0.99) | 0.60 (0.59–0.61) |
| | 55–64 | 0.54 (0.52–0.56) | 0.83 (0.82–0.84) | 0.16 (0.15–0.17) | 0.99 (0.98–0.99) | 0.61 (0.60–0.63) |
| | 65–74 | 0.57 (0.55–0.59) | 0.84 (0.83–0.86) | 0.16 (0.15–0.17) | 0.99 (0.99–0.99) | 0.62 (0.61–0.64) |
| | 75+ | 0.61 (0.58–0.63) | 0.85 (0.84–0.87) | 0.17 (0.15–0.18) | 0.99 (0.99–0.99) | 0.63 (0.61–0.65) |
| Education level | None | 0.30 (0.24–0.35) | 0.73 (0.67–0.78) | 0.14 (0.12–0.16) | 0.95 (0.92–0.97) | 0.55 (0.51–0.60) |
| | Primary | 0.35 (0.32–0.39) | 0.75 (0.72–0.79) | 0.14 (0.13–0.16) | 0.96 (0.95–0.98) | 0.57 (0.54–0.60) |
| | Secondary | 0.41 (0.39–0.43) | 0.78 (0.77–0.80) | 0.15 (0.14–0.16) | 0.98 (0.97–0.98) | 0.58 (0.56–0.60) |
| | Tertiary | 0.48 (0.47–0.49) | 0.81 (0.80–0.82) | 0.15 (0.15–0.16) | 0.98 (0.98–0.99) | 0.60 (0.58–0.60) |

(*Continued*)

**Table 2.** (Continued)

| Predictor variable | | Model 1 | Model 2 | Model 3 | Model 4 | Model 5 |
|---|---|---|---|---|---|---|
| | | Probability of neutering (95% CI) | Probability of answering *Never allow dog to roam* (95% CI) | Probability of answering *Strongly agree* to question "*I do not like the presence of free-roaming dogs around home or work*" (95% CI) | Probability of answering *Yes* to the question "*Should an increase in free-roaming dogs be prevented*" (95% CI) | Probability of answering *No free-roaming dogs* to question "*Would you prefer to see*"(95% CI) |
| *Threatened by dogs on the street* | Strongly disagree | NI | NI | **0.05 (0.047–0.054)** | **0.97 (0.97–0.98)** | **0.49 (0.48–0.50)** |
| | Disagree | NI | NI | **0.13 (0.12–0.13)** | **0.98 (0.98–0.98)** | **0.57 (0.56–0.58)** |
| | Neutral | NI | NI | **0.26 (0.25–0.27)** | **0.99 (0.99–0.99)** | **0.65 (0.64–0.66)** |
| | Agree | NI | NI | **0.44 (0.43–0.45)** | **0.99 (0.99–0.99)** | **0.73 (0.72–0.74)** |
| | Strongly agree | NI | NI | **0.64 (0.62–0.65)** | **1.00 (0.99–1.00)** | **0.79 (0.78–0.81)** |
| *Country* | Bulgaria | **0.41 (0.39–0.43)** | **0.61 (0.59–0.62)** | **0.20 (0.18–0.21)** | 0.99 (0.98–0.99) | **0.55 (0.53–0.56)** |
| | Italy | **0.62 (0.60–0.64)** | **0.92 (0.91–0.93** | **0.10 (0.10–0.11)** | 0.98 (0.98–0.99) | **0.76 (0.74–0.77)** |
| | Ukraine | **0.37 (0.36–0.38)** | **0.81 (0.81–0.82)** | **0.17 (0.16–0.17)** | 0.98 (0.98–0.98) | **(0.44–0.46)** |

**Significant results are highlighted in bold.** NI = predictor variable not included in the model.

uneducated had a lower probability of neutering and a higher probability of allowing dogs to roam. Respondents who identified as: female, feeling threatened by free-roaming dogs, older, or having more education had a higher probability of answering that increases in free-roaming dog numbers should be prevented. This information could be used to target interventions towards groups who are at-risk of irresponsible dog ownership behaviours [20, 26].

## Ownership practices

Responsible ownership is an important component of dog population management [19]. In order to effectively target dog population management interventions, it is important to understand the actions taken by dog owners to acquire, provide care, and relinquish ownership. Most respondents in Italy reported that they acquired their dogs from a shelter, whereas in Bulgaria and Ukraine most respondents reported acquiring their dogs from friends or by finding a dog in the street. The differences in dog acquiring behaviour could be due to a lack of public awareness of local shelters, or perceived differences in shelter quality between the study countries. However, there is currently little research to substantiate these explanations and more work on public awareness is needed. In all study countries, many respondents had adopted a dog directly from the street, potentially reflecting the prevalence of free-roaming dogs in these countries. This may also provide an explanation for the lower uptake from shelters. Where free-roaming dogs are prevalent, people may easily adopt dogs from streets near their homes, rather than travelling to a shelter to adopt a dog. Fewer respondents in Italy paid for their dog. Previous studies have suggested that dogs who are received for little cost are at higher risk of relinquishment [37]. However, the number of respondents who answered that they had given up a dog was low across the study countries. These numbers are likely to be an underestimate, given the taboo around relinquishing dogs. A study by Hsu, Severinghaus and Serpell (2003) [27] found similar estimates, where only 5.3% of respondents answered that they had given up a dog, though far more respondents answered that they knew someone who had given up a dog (31.9%). This indicates that respondents may underreport relinquishment

of owned dogs. Additionally, the self-selection process of recruitment for this questionnaire may result in respondents who are more highly engaged with their dog and dog ownership and less likely to relinquish their dogs.

Responsible dog ownership requires that an owner provides care for a dog until it dies or is transferred to another owner [19]. Most respondents who had relinquished a dog in Italy and Ukraine reported they had given their dog to a friend (Fig 1), complying with responsible ownership [19]. In Bulgaria, a higher percentage of respondents answered they had "*Let free*" their dog (Fig 1). Letting a dog free to the street directly increases the free-roaming dog population. Previous studies have found that respondents prefer to let a dog free to the street as it offers the dog an opportunity to live, unrestricted, outside of a shelter and offers the possibility to find another owner through adoption from the street [27]. This suggests that some dog owners may perceive letting a dog free to the street as responsible ownership. Further research is required to understand why respondents in Bulgaria chose to let a dog free, instead of giving to a shelter or to another owner.

Preventing the production of unwanted puppies is an important part of responsible ownership [19]. Most respondents answered that they prevented their dogs from reproducing; 50.8% of respondents in Bulgaria, 65.3% in Italy, and 35.3% in Ukraine answered that they did so through neutering. This compares to study sample neutering percentages of 54% in the United Kingdom [38], and up to 80% in Australia [39, 40]. The results of the present study should be interpreted with caution, as the self-selection process of recruiting for questionnaires can result in biased samples of the populations. It is possible that respondents who were more likely to neuter their dogs (such as those with higher levels of education) were more likely to complete the questionnaire. The true proportion of neutered owned dogs in the study countries may be lower. Neutering of owned dogs can prevent unwanted offspring and, if owned dogs are free-roaming, can help to prevent unowned dogs from reproducing. When respondents were asked why they did not neuter their dog, the most common answer (if one was provided) across all countries was that a dog should reproduce at least once (Fig 1). Few respondents answered that it was for cost reasons. This contrasts with previous findings in Taiwan [27] and Brazil [41], where respondents cite cost and "too much trouble" as primary reasons for not neutering. As cost, in this study, was not found to be a primary reason that owners did not neuter their dogs, this suggests that in Bulgaria, Italy, and Ukraine, whilst low-cost or free neutering interventions may be important [42], interventions should also address owner knowledge, attitudes, and practices towards reproduction, in order for interventions to have a greater impact.

Our findings of significant associations between country, gender, religious belief, reason for ownership, age, and education level and the probability of neutering (S9 Table) reflect those reported in other studies [20–24, 26]. For example, a study by Fielding (2007) in New Providence, The Bahamas [21] also found that respondents with higher levels of education were more likely to have neutered their dog. Similarly, Costa *et al*., (2017) [26] found that respondents with higher levels of education were more likely to answer that neutering was the best way to control the overabundance of free-roaming animals in Brazil. Respondents with higher levels of education may have a higher level of awareness of responsible ownership and the benefits of neutering, in addition to potentially having a higher income and ability to pay for neutering. Lower education levels may mean less knowledge of possible effects (and lack thereof) of reproducing on a dog's health and behaviour. Fielding, Samuels & Mather (2002) [20] also found significant associations between owner age and neutering probability, suggesting that younger owners may have a greater desire to breed from their dog, compared to older owners. These findings suggest interventions could be targeted towards younger owners and those with a lower level of education to increase knowledge of the possible effects of neutering and awareness of responsible ownership practices.

Owned dogs that are free-roaming directly increase the size of the free-roaming population. Owned free-roaming dogs therefore contribute to the issues, such as the risks to public health [43, 44] and wildlife [11, 12, 45–49]. Efforts encouraging responsible ownership may help reduce the number of dogs roaming, and may therefore help to reduce the impacts of the free-roaming dogs on public health and wildlife [19]. This study found evidence for significant associations between gender, religious beliefs, reason for dog ownership, age, education, and country, and the probability of allowing owned dogs to roam (S9 Table). It is clear from these results that interventions should be targeted using these demographic risk factors to prevent roaming behaviour, particularly in countries where higher percentages of owned dogs are free-roaming, such as Bulgaria and Ukraine.

Most respondents (59–92%) across all three countries answered that they never allowed their dogs to roam (S5 Table). Again, these results might be biased, as more highly engaged dog owners may have been more likely to participate in this study and may also be less likely to allow their dogs to roam. The results are higher than those reported in studies using similar sampling approaches (i.e. relying on voluntary participation in questionnaires/interviews) in the Bahamas 57% [50], Bhutan 50% [51], Cameroon 37.7% [52], Guatemala 25.7% [53], urban households in Haiti 54% [54], Kenya 19% [55], Mexico 44.9% [56], Ethiopia 15.7% [57], Tanzania 22% [58], and Uganda 21.7% [59], but lower than those reported in semi-urban households in Haiti 62% [54] and Taiwan 79% [27]. There was a significant association between study country and roaming probability (S9 Table), with respondents in Bulgaria more likely to allow their dogs to roam, compared to Italy and Ukraine. The significant association with country may reflect differences in dog ownership practices and attitudes between the study countries.

## Attitudes towards free-roaming dogs

In Bulgaria and Ukraine, almost all respondents answered that they had seen free-roaming dogs on the street, whereas in Italy 18.7% of respondents had never seen a free-roaming dog (Fig 2). These results may indicate that the populations of free-roaming dogs are larger in Bulgaria and Ukraine. Within Italy, there are differences in dog population management: some regions permit CNR and the presence of "community dogs" (free-roaming dogs owned by the municipality), whilst other regions only permit dog population management through sheltering. Respondents living in regions that do not permit community dogs, or in regions with smaller free-roaming dog populations, may be expected to observe fewer free-roaming dogs. Higher percentages of respondents in Bulgaria and Ukraine answered that they felt threatened by free-roaming dogs, and that they or a member of their family had been bitten in the last 12 months. These results may also indicate a greater free-roaming dog population size and related problems in Bulgaria and Ukraine.

A large proportion of respondents across all countries answered that they provided care for free-roaming dogs (S6 Table). For example, 90.6% in Bulgaria, 53.7% in Italy, and 67.5% of respondents in Ukraine answered that they provided food for free-roaming dogs. For Bulgaria and Ukraine, these numbers are similar to those reported by Costa et al., (2015) in Brazil, where 61.9% of respondents reported that they or their neighbours fed free-roaming animals, and Massei et al., (2017) [60] in Nepal, where 47% of respondents provided food and care for free-roaming dogs. In a previous study by Slater et al., (2008) [61] in central Italy, only 5% of respondents reported that they provided care for free-roaming dogs. This is much lower than the numbers reported in this study, where 71.5% of Italian respondents answered that they provided care for free-roaming dogs. This may be explained by the potential bias in the recruitment process of this study, respondents who provide care for free-roaming dogs may also have been more motivated to complete the questionnaire. Data was collected by Slater et al., (2008)

using an anonymous telephone survey and had a high response rate (74%). Providing care for free-roaming dogs is controversial. Providing food may alleviate welfare issues associated with lack of nutrition in the free-roaming dog population [62–64], but also increases the carrying capacity for the free-roaming dog population.

Most respondents across all study countries felt that the municipal government and volunteer organisations should be responsible for managing free roaming dog populations, and mostly by methods such as sheltering, CNR, and by controlling the breeding of owned dogs (Fig 3). These results are similar to those found in previous studies [61, 65, 66]. For example, a study by Ortega-Pacheco et al., (2007) [66] in Yucatan, Mexico found that 52.8% of interviewed households supported the neutering of dogs for dog population management, and felt that the government and society were responsible for dog population management. The results in this study suggest there is support for dog population management through sheltering, CNR, and restricted breeding of owned dogs. Few respondents answered that culling should be used to control the free-roaming dog population (Fig 3). These results are similar to those found by Beckman et al., (2014) [65], but are much lower than results by Costa et al., (2017) [26], where culling was supported by 26.8% of respondents.

As attitudes can play an important role in determining the success of dog population management, it is important that organisations involved in dog population management gauge the level of support for reducing free-roaming dogs in the area. Across all three countries, most respondents answered that they would prefer to see fewer or no free-roaming dogs, and that an increase in free-roaming dogs should be prevented. With regards to Italy, these responses correspond with previously reported attitudes in the Teramo province in the Abruzzo region of Italy [61].

## Implications for future interventions

The results of this study suggest that there is a preference in all three study countries for a reduction in free-roaming dog numbers, and for this to be achieved through sheltering, CNR and responsible ownership, rather than culling. There is therefore support for the management interventions that are taking place in these study countries. Targeted interventions that can influence the behaviour of those less likely to practice responsible ownership may help to improve responsible ownership and reduce free-roaming dog numbers. For example, as there was evidence for significant associations between gender and age on response variables roaming and neutering, interventions could be adapted to target men and younger people on responsible ownership practices. For example, interventions could target these groups to increase knowledge of the necessity to neuter dogs and possible effects of neutering on dog health and behaviour.

Questionnaires are important tools for evaluating the impact of interventions on human attitudes and dog ownership practices. This includes monitoring attitudes and behaviour (such as responsible ownership) to determine whether education campaigns are having a significant impact. There have been numerous studies on attitudes towards free-roaming dogs and dog ownership practices, but few repeated surveys to assess the effectiveness of dog population control on human attitudes and dog ownership practices [26, 67]. The results from this present study can be used to target interventions to those who are less likely to practice responsible ownership and the results can also be used as a baseline for monitoring the effect of dog population management interventions on dog ownership behaviours and attitudes in Bulgaria, Italy, and Ukraine.

## Limitations of questionnaire research methods

As discussed throughout, there are limitations in using questionnaires to determine attitudes and behaviours. The self-selection process involved in the recruitment for questionnaires can

result in a biased sample of the target population, as certain members of the public may be more motivated to complete the questionnaire, for example dog owners, or those with strong views about the subject. In this questionnaire, as with other similarly themed questionnaire [61], a high percentage of the respondents were female. As there were fewer responses from men, the questionnaire results may not necessarily reflect the views of the wider population. A high percentage of respondents reported to have or be in tertiary education, which is not representative of the wider populations. The questionnaire was also primarily advertised through social media; therefore, members of the public who do not have access to social media are likely to have been missed. Although this is a limitation, social media provides opportunities to recruit a large and diverse range of respondents (see [68, 69] for review). We used four different adverts with slightly different wording in order to attract as many respondents as possible. The different adverts may have attracted different subsets of people, which could lead to biases in results. Though we were unable to determine which adverts respondents saw, all adverts were distributed equally across the three countries and are unlikely to lead to differences between countries.

In questionnaire surveys, missing data can occur at two levels: (i) missing data of the complete questionnaire (as described above), and (ii) missing data when respondents do not complete specific questions or sections. In this study, higher percentages of missing responses were observed (S1 Table) for questions relating to the respondents' religion (13.3%), whether they did not like the presence of free-roaming dogs (10.2%), and if they felt threatened by free-roaming dogs around their home or work (8.2%). This missing data can result in biased estimates, and as such, these results should be interpreted with caution. Despite the clear biases in questionnaire surveys, given the range of respondents in this study (for example, in terms of ages and regions), the results provide an indication of ownership practices and attitudes, and the statistical models still give us information about the risk factors for behaviours and attitudes.

## Conclusions

When planning dog population management interventions, it is important to understand how human behaviour may impact the success of an intervention. This involves understanding how dog ownership practices may influence intervention success, and gauging the level of public support for management interventions. This study found evidence for significant associations between demographic factors and ownership practices and respondent attitudes. These results can be used to inform future dog population management interventions in these countries. Interventions should consider also carrying out periodic questionnaire surveys to evaluate changes in respondent attitudes towards responsible ownership and the free-roaming dog population.

## Supporting information

**S1 File. Facebook adverts.**
(DOCX)

**S2 File. English copy of questionnaire.**
(DOCX)

**S3 File. Answer option to question "*Are your dog(s) registered and identified*" in Bulgarian questionnaire.**
(DOCX)

**S1 Table. Number of "No responses" to outcome and predictor variables in statistical analysis.**
(DOCX)

**S2 Table. Demographic information about respondents in Bulgaria, Italy and Ukraine.**
(DOCX)

**S3 Table. Number of respondents in Bulgaria, split by oblasts in Bulgaria.**
(DOCX)

**S4 Table. Number of respondents in Italy, split by regions in Italy.**
(DOCX)

**S5 Table. Number of respondents in Ukraine, split by oblasts in Ukraine.**
(DOCX)

**S6 Table. Respondents answers to questions about ownership practices in Bulgaria, Italy and Ukraine.**
(DOCX)

**S7 Table. Respondents answers to questions about attitudes to free-roaming dogs in Bulgaria, Italy and Ukraine.**
(DOCX)

**S8 Table. The posterior mean values, error estimates, the 2.5 and 97.5 percentiles of the posterior distribution (CI), Rhat values and bulk and tail effective sample sizes (ESS) for Model 1.**
(DOCX)

**S9 Table. The posterior mean values, error estimates, the 2.5 and 97.5 percentiles of the posterior distribution (CI), Rhat values and bulk and tail effective sample sizes (ESS) for Model 2.**
(DOCX)

**S10 Table. The posterior mean values, error estimates, the 2.5 and 97.5 percentiles of the posterior distribution (CI), Rhat values and bulk and tail effective sample sizes (ESS) for Model 3.**
(DOCX)

**S11 Table. The posterior mean values, error estimates, the 2.5 and 97.5 percentiles of the posterior distribution (CI), Rhat values and bulk and tail effective sample sizes (ESS) for Model 4.**
(DOCX)

**S12 Table. The posterior mean values, error estimates, the 2.5 and 97.5 percentiles of the posterior distribution (CI), Rhat values and bulk and tail effective sample sizes (ESS) for Model 5.**
(DOCX)

## Acknowledgments

We thank Sarah Ross, Benjamin Cueni, Alesya Lischyshyna, Greta Berteselli, and Matteo Chincarini for providing support with Facebook advertising and translation and all survey respondents for contributing to the study. We are also grateful to the reviewers for providing useful feedback on this paper, and to Dr Helen Gray, Dr Conor Goold, and Dr Mary Friel for reviewing draft versions and providing important feedback on statistical methods.

## Author Contributions

**Conceptualization:** Lauren Margaret Smith, Rupert Quinnell, Alexandru Munteanu, Sabine Hartmann, Paolo Dalla Villa, Lisa Collins.

**Data curation:** Lauren Margaret Smith.

**Formal analysis:** Lauren Margaret Smith.

**Methodology:** Lauren Margaret Smith, Rupert Quinnell, Lisa Collins.

**Supervision:** Rupert Quinnell, Lisa Collins.

**Writing – original draft:** Lauren Margaret Smith.

**Writing – review & editing:** Lauren Margaret Smith, Rupert Quinnell, Alexandru Munteanu, Sabine Hartmann, Paolo Dalla Villa, Lisa Collins.

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
