## [Decision Letter · Decision Letter 0]

8 Sep 2021

PONE-D-21-15770Public attitudes towards free-roaming dogs and dog ownership practices in Bulgaria, Italy, and UkrainePLOS ONE

Dear Dr. Smith,

Thank you for submitting your manuscript to PLOS ONE. After careful consideration, we feel that it has merit but does not fully meet PLOS ONE’s publication criteria as it currently stands. Therefore, we invite you to submit a revised version of the manuscript that addresses the points raised during the review process.

For details please see the reviewer reports below.

We look forward to receiving your revised manuscript.

Kind regards,

I Anna S Olsson, Ph.D.

Academic Editor

PLOS ONE

Journal Requirements:

3. Thank you for stating the following in the Competing Interests section: "The authors declare that: A.M.M. and S.H. are employed by VIER PFOTEN International, a global animal welfare organisation; L.M.C has received a research grant from VIER PFOTEN International; and L.M.S.’s research has been funded by VIER PFOTEN International." 

Reviewers' comments:

Reviewer's Responses to Questions

**Comments to the Author**

1. Is the manuscript technically sound, and do the data support the conclusions?

Reviewer #1: Yes

Reviewer #2: Partly

2. Has the statistical analysis been performed appropriately and rigorously? 

Reviewer #1: Yes

Reviewer #2: Yes

3. Have the authors made all data underlying the findings in their manuscript fully available?

Reviewer #1: Yes

Reviewer #2: Yes

4. Is the manuscript presented in an intelligible fashion and written in standard English?

Reviewer #1: Yes

Reviewer #2: Yes

5. Review Comments to the Author

Reviewer #1: With much interest I have read and reviewed the manuscript on the study of ‘Public attitudes towards free-roaming dogs and dog ownership practices in Bulgaria, Italy, and Ukraine’. The study discusses 1) feelings, thoughts and preferences regarding stray/free roaming dogs, which are studied as attitudes towards these dogs, 2) the practice of neutering owned dogs and allowing owned dogs to roam, which are studied as dog ownership practices. The study also determines 3) dog acquisition and abandonment practices. Data was collected in three countries, descriptive data is presented and the three topics are studied for associations with demographic variables, including country of origin of the respondent and for associations with for instance feelings of fear for stray/free roaming dogs.

I would recommend the study to be accepted for submission by PloSOne and feel the study has value as it covers a study area that is of much relevance to dog welfare and the animal-human bond. A strong point of the study is that it was done in three countries, allowing for comparison between these countries’ respondents. Yet, the manuscript would benefit from some improvements, for which I would suggest:

Abstract

• Would you have enough words left to add information on the study method? The abstract now moves from aim (line 21) directly to results (line 23).

• Line 27 indicates associations between ‘public attitudes and dog ownership practices’. However, is it not 1) associations between attitudes and gender (…) and 2) associations between dog ownership practices and gender (…) that you studied?

• Line 30 and 33: are the studied variables jointly leading to higher probabilities or separately? You now use the word ‘and’. Would ‘or’ be a better choice?

• Please consider a different final sentence for the abstract. The current choice suggests something that was not studied and working cohesively towards a shared goal may require another approach than stated here.

Introduction

• The final sentence of the first paragraph (line 48-50) doesn’t seem to connect this paragraph to the next. It is not the assessment of management programs that is key to your study, right? The next paragraph is on public attitudes and their role in dog population (management). Therefore, it may help the flow of your text to stay with your topics and to choose a connecting sentence with content on these topics. (As to avoid that the reader expects the next paragraph to provide details on assessment of population management). A similar disruption of text flow is in line 64-65. Here ‘Reduction in numbers (…) disease control (20-22)’, hints upon the (out of scope) topic of carrying capacity. However, you wish to discuss the role of public attitudes. Perhaps rephrase to something like: ‘Without a change in demand for dogs in the community, new dogs may be bought, adopted or not prevented from moving into a community. The latter shift in dog populations may be consequential to the community’s habitat offering food, shelter, and/or social requirements’. (And check if the references still apply.)

• In line with the suggestions on terminology below, your readers will be helped by indicating somewhere in the introduction how you define/study ‘attitude’ and ‘ownership practices’ in this study, as both concepts theoretically cover a broad range of factors.

Consistency in terminology and choice thereof

• Please check the manuscript for consistency in terminology and choice thereof. For example, both ‘free-roaming dogs’ and ‘stray’ are used.

• Another example is the use of the word ‘social factors’ in line 37-38 of the abstract, however, is this a correct reflection of the factors that you studied?

• In the introduction: ‘strategies’ or ‘intervention (programme)s’, or both? (line 54 and 56)

• Is the OIE quote formulated as a definition? ‘there should be’ indicates that it is a list of requirements, not a definition?

• Are ‘dog abandonment level and reasons’ dog ownership practices? Are ‘acquisition reasons and manners’ dog ownership practices?

• Line 484: new terminology of ‘culture’: do you need to introduce this new terminology here? Of can you stay closer to one of the terminologies/definitions/concepts previously used/addressed in your study?

Study sample

• Would it be possible to add early on a clarification on the study sample: was it one study sample that was questioned on both topics (attitudes versus ownership practices)? From S2 Table 2 I would conclude that you used one sample and for some questions only the responses of the dog-owners were used. The clarification of variables (eg Table 1) provides the answer in the levels of dog owner versus non-dog owner. However, an earlier clarification would help the reader and please do consider to include the ratio. P.s. You may even reconsider using the word ‘public’ in ‘public attitudes’ in the title. The nature of the study may have attracted only a certain part of ‘the general public’. The study approach of social media targeting may also have affected who is in the sample. Like many studies (on dogs) the female-male ratio is disbalanced in this study also. These matters were discussed nicely in lines 558 onwards, and you may want to opt to adjust the title accordingly.

Data and Methods

• Table 1 indicates percentages of >7.5% missing values for ‘religion’, ‘do not like stray presence’, ‘feel threatened’, would you want to comment on this in the discussion, particularly on the latter two?

Results

• The descriptive analyses are given in percentages with the numbers of overall respondents in line 206. This facilitates readability of the text, but the downside is less accuracy of the info, which would increase by adding (N=xx of xxx).

• In the statistical analyses section subsections are titled with ‘effect of…’ this is correct as it refers to the statistical method/model used, however it may unintentionally suggest a cause-effect relationship to less experienced readers. As PloSOne is open access, you may want to consider to change the titles of the subsections. A minor thing in these titles is that there is some inconsistency in ‘on answer to’, ‘on answering’, ‘on the question’. Shorter, more aligned formatting will help the reader. Please also look at the title on page 18 again: is ‘respondent experience’ precise enough for readers to know what the section is about (and note the dot at the end of the title).

• In Table 2 the description under Model 3 would benefit from adding the topic (of stray dog presence), as was done in the other cells of this row. Some of the lower listed probabilities in this Table have three decimal places, whereas above these all have two (eg final column for children in household ‘0.593’ and see in particular Model 4 – age and below There also seems to be an indication ‘t0’ in column Model 3, 6th row from below).

Discussion

• The first paragraph of the Discussion section on Ownership practices seems to address elements not studied. For instance ‘feeding and vaccination’. Also, was ‘quantifying the prevalence of ownership practices’ key of what you did? In addition: would you not want to mention the attitudes here also?

• The discussion could be more strongly structured. You could opt for a structure of summing up key results and then discussing the findings, after which you discuss limitation. Alternatively, you could present a key result and discuss it. You could also look at stronger starts of each paragraph. For instance, line 402 now starts with ‘There were (…)’ and line 450 with ‘This study found evidence’: such sentence parts make your writing less concise. Consequentially, it is harder for the reader to find the flow or logic between the paragraphs.

• Regarding the content of the discussion:

- Line 404 onwards indicates ‘a lack of public awareness of local shelters’, but a few sentences onwards you indicate how availability of dogs on the streets may affect ‘acquisition’. Yet, without indicating that this could also be an explanation for lesser uptake from shelters, that is if dogs are available more so ‘close by on the street’, why go to a shelter?

- Line 411 re. ‘low number of giving up’: well done that you cite the literature and possible underreporting. An addition could be to reflect a little on how your approach to gather participants may have impacted on your results? (Could these participants have been more highly engaged with their dog or with dog ownership?) You do touch upon this in lines 431 onwards and there is a nice section on it from line 500 onwards, so perhaps by restructuring the Discussion this comment becomes superfluous.

- With regard to line 411 onwards, there are some references on neutering percentages in study samples. It may put your data in perspective? (See for instance Diesel G, Brodbelt D, Laurence C. Survey of veterinary practice policies and opinions on neutering dogs. Vet Rec Open. 2010;166(15):455-458. & Bennett PC, Rohlf VI. Owner-companion dog interactions: Relationships between demographic variables, potentially problematic behaviours, training engagement and shared activities. Appl Anim Behav Sci. 2007;102(1-2):65-84. And this ref may also be of interest: Rohlf VI, Bennett, PC, Toukhsati S, Coleman G. Why do even committed dog owners fail to comply with some responsible ownership practices? Anthrozoös. 2010;23(2):143-155.)

- Lines 450-461: and how about your study’s finding of owners thinking ‘a dog should reproduce at least once’: could lower education levels also mean lesser knowledge of possible effects (and lack thereof) of reproducing on a dog’s health and behaviour? You may have a nice opportunity here to write about attitude, knowledge, skills and behaviour and argument the statement you make later how attitudes need to be addressed as well when aiming for effective dog population management.

- Line 463 onwards: would you not want to link with mental health (next to health)? You can discuss your findings on respondents that were bitten/had family bitten and not liking the presence of stray dogs, which may indicate mental health risks for these respondents?

- Line 476: yes, and why would you anticipate this lower likeliness? Some brief words on more highly engaged dog owners being more likely to partake in studies on dogs (as discussed well in the limitation section)? Well done that you indicate percentages from other studies for comparison: an addition could be to indicate whether the study approaches differed from yours?

- Lines 538-545: You end this interesting paragraph with age as a targeting option. Would you not want to add how the attitude aspect that associated neutering in your study could be addressed specifically?

- Line 548: Mind that questionnaires do not measure on behaviour itself.

Further suggestions

- Abstract (line 20): ‘behaviour and outlook’, but your aim (one sentence lower) regards attitudes of dog owners and non-dog owners & practices of dog owners. Is ‘behaviour and outlook of local communities’ the right step up to your study aim? (Also in line 94 of the Introduction).

- Abstract (line 23 & 26): This study – this questionnaire: you may want to reword it (esthetical reasons only).

- Abstract (line 35): Can your findings help to ‘inform’ or ‘shape’ management interventions?

- Introduction (line 61): ‘these factors’: is it clear to the reader which factors?

- Introduction (line 83): ‘encouraging’ – but note that legislation may have an effect other than or next to an encouraging effect.

- If available, indications of ‘other reasons’ (eg Results line 239) would be informative when percentages are high.

Reviewer #2: Thank you for the opportunity to review this manuscript that details dog ownership behaviors and attitudes towards free-roaming dogs in Bulgaria, Italy, and Ukraine. The study employed an online questionnaire and recruited participants via social media. The data showed differences in self-reported behaviors and attitudes between countries, and found statistically significant associations between demographic factors such as age, gender, education, and the measured outcomes.

The submitted article satisfies PLOS ONE’s 7 publication criteria. It is very well written and technically sound, and the limitations are adequately addressed in the discussion. There are a few instances in which the authors draw conclusions or make statements that go beyond the reach of their data, but this can be easily fixed with minor editing.

Going beyond the data: At several points in the manuscript the authors discuss the efficacy of free-roaming dog population control (e.g., abstract, lines 37, 95), but efficacy was not assessed in their study. They surveyed attitudes and behaviors, then hypothesized how those behaviors might affect efficacy, but they did not assess it as they claim.

Causality is sometimes implied, but causal relationships cannot be determined from the questionnaire data. This is true when the authors describe the relationship between demographic data and attitudes and behavior: I would caution against saying there was a “significant effect of gender/age/education/feeling threatened” on response variables because cause and effect were not assessed. Instead, I suggest saying there was a statistically significant association between gender/age/etc. and the response variable.

I have concerns about the advertisements used for recruitment. The four advertisements are very different, and depending on which advertisement a participant saw, their responses may have been biased. For example, Advert 3 includes the language “stray dog overpopulation is a global problem which is of public health, animal welfare and environmental concern,” whereas Advert 1 simply states that they want a wide range of opinions on stray dogs. This is problematic and needs to be addressed in the methods and/or discussion. How were these advertisements distributed? Was Advert 1 used in one country or in one language while Advert 3 was used in another? If so, significant differences cannot necessarily be attributed to region as they could instead be attributed to the condition of the advertisement. If they were randomly distributed across country/language then it may not be a confound, but there could still be an effect of advertisement and this should be discussed.

Line 119 – do you mean Power analysis (instead of sample size)?

6. PLOS authors have the option to publish the peer review history of their article (what does this mean?). If published, this will include your full peer review and any attached files.

Reviewer #1: **Yes: **I.R. van Herwijnen

Reviewer #2: No

---

## [Author Response · Author response to Decision Letter 0]

15 Nov 2021

Dear Editor,

We would like to thank both you and the reviewers for your helpful feedback on our manuscript. We have addressed each of the comments, as outlined below.

All line numbers below refer to the revised manuscript with track changes (full markup).

Reviewer 1

With much interest I have read and reviewed the manuscript on the study of ‘Public attitudes towards free-roaming dogs and dog ownership practices in Bulgaria, Italy, and Ukraine’. The study discusses 1) feelings, thoughts and preferences regarding stray/free roaming dogs, which are studied as attitudes towards these dogs, 2) the practice of neutering owned dogs and allowing owned dogs to roam, which are studied as dog ownership practices. The study also determines 3) dog acquisition and abandonment practices. Data was collected in three countries, descriptive data is presented and the three topics are studied for associations with demographic variables, including country of origin of the respondent and for associations with for instance feelings of fear for stray/free roaming dogs. 

I would recommend the study to be accepted for submission by PloSOne and feel the study has value as it covers a study area that is of much relevance to dog welfare and the animal-human bond. A strong point of the study is that it was done in three countries, allowing for comparison between these countries’ respondents. Yet, the manuscript would benefit from some improvements, for which I would suggest: 

Abstract

• Would you have enough words left to add information on the study method? The abstract now moves from aim (line 21) directly to results (line 23).

Thank you for this suggestion – we’ve included details on the questionnaire and statistical analysis in the abstract now (lines 25-29).

• Line 27 indicates associations between ‘public attitudes and dog ownership practices’. However, is it not 1) associations between attitudes and gender (…) and 2) associations between dog ownership practices and gender (…) that you studied?

That’s correct – we’ve amended to “We identified significant associations between both attitudes and ownership practices with…”

• Line 30 and 33: are the studied variables jointly leading to higher probabilities or separately? You now use the word ‘and’. Would ‘or’ be a better choice?

The variables each separately lead to higher probabilities, so we’ve changed “and” to “or” as suggested.

• Please consider a different final sentence for the abstract. The current choice suggests something that was not studied and working cohesively towards a shared goal may require another approach than stated here.

We’ve omitted this statement.

Introduction

• The final sentence of the first paragraph (line 48-50) doesn’t seem to connect this paragraph to the next. It is not the assessment of management programs that is key to your study, right? The next paragraph is on public attitudes and their role in dog population (management). Therefore, it may help the flow of your text to stay with your topics and to choose a connecting sentence with content on these topics. (As to avoid that the reader expects the next paragraph to provide details on assessment of population management). A similar disruption of text flow is in line 64-65. Here ‘Reduction in numbers (…) disease control (20-22)’, hints upon the (out of scope) topic of carrying capacity. However, you wish to discuss the role of public attitudes. Perhaps rephrase to something like: ‘Without a change in demand for dogs in the community, new dogs may be bought, adopted or not prevented from moving into a community. The latter shift in dog populations may be consequential to the community’s habitat offering food, shelter, and/or social requirements’. (And check if the references still apply.)

Thank you for these suggestions, we have omitted the final sentence in the first paragraph and included a sentence, based on your suggestion:

“Unless this demand reduces, new dogs may be acquired to replace those removed by population management (either bought, adopted, or by uncontrolled immigration of free-roaming dogs)” Lines 72-74.

• In line with the suggestions on terminology below, your readers will be helped by indicating somewhere in the introduction how you define/study ‘attitude’ and ‘ownership practices’ in this study, as both concepts theoretically cover a broad range of factors. 

We have included the definitions:

“In this study, we define attitudes as the thoughts, feelings, and opinions of respondents, as reported in the questionnaire. We define ownership practices as the actions taken to acquire, provide care, and relinquish ownership of dogs, as reported in the questionnaire.” (lines 108-111).

Consistency in terminology and choice thereof

• Please check the manuscript for consistency in terminology and choice thereof. For example, both ‘free-roaming dogs’ and ‘stray’ are used. 

We have changed all to “free-roaming dogs” for accuracy and consistency.

• Another example is the use of the word ‘social factors’ in line 37-38 of the abstract, however, is this a correct reflection of the factors that you studied?

Now omitted from the abstract.

• In the introduction: ‘strategies’ or ‘intervention (programme)s’, or both? (line 54 and 56)

We have changed all occurrences of “strategies” to “interventions”.

• Is the OIE quote formulated as a definition? ‘there should be’ indicates that it is a list of requirements, not a definition?

Thank you for this – we have changed to “The OIE describes the requirements of responsible ownership as” (line 84).

• Are ‘dog abandonment level and reasons’ dog ownership practices? Are ‘acquisition reasons and manners’ dog ownership practices?

We have amended to “determine local ownership practices and attitudes” (line 113).

• Line 484: new terminology of ‘culture’: do you need to introduce this new terminology here? Of can you stay closer to one of the terminologies/definitions/concepts previously used/addressed in your study?

We have amended to “The significant effect of country may reflect differences in dog ownership practices and attitudes between the study countries.”.

Study sample

• Would it be possible to add early on a clarification on the study sample: was it one study sample that was questioned on both topics (attitudes versus ownership practices)? From S2 Table 2 I would conclude that you used one sample and for some questions only the responses of the dog-owners were used. The clarification of variables (eg Table 1) provides the answer in the levels of dog owner versus non-dog owner. However, an earlier clarification would help the reader and please do consider to include the ratio. P.s. You may even reconsider using the word ‘public’ in ‘public attitudes’ in the title. The nature of the study may have attracted only a certain part of ‘the general public’. The study approach of social media targeting may also have affected who is in the sample. Like many studies (on dogs) the female-male ratio is disbalanced in this study also. These matters were discussed nicely in lines 558 onwards, and you may want to opt to adjust the title accordingly. 

Thank you for this suggestion. We agree and have omitted “public” in the title as suggested. 

We have amended the section “Questionnaire design” to include:

“The questionnaire comprised closed questions regarding the respondents’ attitudes towards free-roaming dogs and their management. Respondents that reported owning a dog were asked to also complete questions relating to dog ownership practices.”(lines 158-161)

And

“ The questionnaire consisted of three sections: (i) socio-demographic information of the respondent (all respondents); (ii) ownership practices (only dog owners); and (iii) attitudes towards the presence of free-roaming dogs and the management of the free-roaming dog population (all respondents)” (lines 162-166)

We have included the number of respondents for this section of the questionnaire in lines 221-222 “Sixty-five percent of respondents in Bulgaria (n=3528), 75% in Italy (n=2581) and 56% in Ukraine (n=10797) reported owning a dog.”

Data and Methods

• Table 1 indicates percentages of >7.5% missing values for ‘religion’, ‘do not like stray presence’, ‘feel threatened’, would you want to comment on this in the discussion, particularly on the latter two?

We have included in the “Limitations of questionnaire research methods” section:

Lines 680-683: “In this study, higher percentages of missing responses were observed (S1 table) for questions relating to the respondents’ religion (13.3%), whether they did not like the presence of free-roaming dogs (10.2%), and if they felt threatened by free-roaming dogs around their home or work (8.2%)”

Results

• The descriptive analyses are given in percentages with the numbers of overall respondents in line 206. This facilitates readability of the text, but the downside is less accuracy of the info, which would increase by adding (N=xx of xxx). 

Added throughout results.

• In the statistical analyses section subsections are titled with ‘effect of…’ this is correct as it refers to the statistical method/model used, however it may unintentionally suggest a cause-effect relationship to less experienced readers. As PloSOne is open access, you may want to consider to change the titles of the subsections. A minor thing in these titles is that there is some inconsistency in ‘on answer to’, ‘on answering’, ‘on the question’. Shorter, more aligned formatting will help the reader. Please also look at the title on page 18 again: is ‘respondent experience’ precise enough for readers to know what the section is about (and note the dot at the end of the title). 

Thank you for this suggestion – we’ve changed the subsections titles and the use of “effect” throughout to prevent this occurring.

• In Table 2 the description under Model 3 would benefit from adding the topic (of stray dog presence), as was done in the other cells of this row. Some of the lower listed probabilities in this Table have three decimal places, whereas above these all have two (eg final column for children in household ‘0.593’ and see in particular Model 4 – age and below There also seems to be an indication ‘t0’ in column Model 3, 6th row from below).

We’ve amended the table to include the suggested column titles and amend the rounding and typo errors.

Discussion

• The first paragraph of the Discussion section on Ownership practices seems to address elements not studied. For instance ‘feeding and vaccination’. Also, was ‘quantifying the prevalence of ownership practices’ key of what you did? In addition: would you not want to mention the attitudes here also?

We have amended the first paragraph of the discussion to read: 

“This study investigated dog ownership practices and attitudes towards the management of free-roaming dogs in Bulgaria, Italy, and Ukraine.”

And, in the second paragraph:

“In order to effectively target dog population management interventions, it is important to understand the actions taken by dog owners to acquire, provide care, and relinquish ownership.”

• The discussion could be more strongly structured. You could opt for a structure of summing up key results and then discussing the findings, after which you discuss limitation. Alternatively, you could present a key result and discuss it. You could also look at stronger starts of each paragraph. For instance, line 402 now starts with ‘There were (…)’ and line 450 with ‘This study found evidence’: such sentence parts make your writing less concise. Consequentially, it is harder for the reader to find the flow or logic between the paragraphs. 

We have amended the discussion structure to improve flow and clarity of logic.

• Regarding the content of the discussion:

- Line 404 onwards indicates ‘a lack of public awareness of local shelters’, but a few sentences onwards you indicate how availability of dogs on the streets may affect ‘acquisition’. Yet, without indicating that this could also be an explanation for lesser uptake from shelters, that is if dogs are available more so ‘close by on the street’, why go to a shelter? 

This is a good point – we’ve included in this paragraph:

“In all study countries, many respondents had adopted a dog directly from the street, potentially reflecting the prevalence of free-roaming dogs in the study countries. This may also provide an explanation for a lesser uptake from shelters. Where free-roaming dogs are prevalent, people may be easily adopt dogs from streets near their homes, rather than having to travel to a shelter to adopt a dog.” (Lines 482-486)

- Line 411 re. ‘low number of giving up’: well done that you cite the literature and possible underreporting. An addition could be to reflect a little on how your approach to gather participants may have impacted on your results? (Could these participants have been more highly engaged with their dog or with dog ownership?) You do touch upon this in lines 431 onwards and there is a nice section on it from line 500 onwards, so perhaps by restructuring the Discussion this comment becomes superfluous. 

We have amended to include:

Lines 494-4956 “Additionally, the self-selection process of recruitment for this questionnaire may result in respondents who are more highly engaged with their dog and dog ownership and less likely to relinquish their dogs.”

- With regard to line 411 onwards, there are some references on neutering percentages in study samples. It may put your data in perspective? (See for instance Diesel G, Brodbelt D, Laurence C. Survey of veterinary practice policies and opinions on neutering dogs. Vet Rec Open. 2010;166(15):455-458. & Bennett PC, Rohlf VI. Owner-companion dog interactions: Relationships between demographic variables, potentially problematic behaviours, training engagement and shared activities. Appl Anim Behav Sci. 2007;102(1-2):65-84. And this ref may also be of interest: Rohlf VI, Bennett, PC, Toukhsati S, Coleman G. Why do even committed dog owners fail to comply with some responsible ownership practices? Anthrozoös. 2010;23(2):143-155.)

Thank you for suggesting these articles. We’ve now included their results for comparison:

Lines 514-515“Most respondents answered that they prevented their dogs from reproducing; 50.8% respondents in Bulgaria, 65.3% in Italy, and 35.3% in Ukraine answered that they did so through neutering. This compares to study sample neutering percentages of 54% in the United Kingdom (36), and up to 80% in Australia (37,38).”

- Lines 450-461: and how about your study’s finding of owners thinking ‘a dog should reproduce at least once’: could lower education levels also mean lesser knowledge of possible effects (and lack thereof) of reproducing on a dog’s health and behaviour? You may have a nice opportunity here to write about attitude, knowledge, skills and behaviour and argument the statement you make later how attitudes need to be addressed as well when aiming for effective dog population management.

Thank you for this suggestion, we have amended this paragraph to include your suggestion and included:

Lines 545-548: “These findings suggest interventions could be targeted towards younger owners and those with a lower level of education to increase knowledge of the possible effects of neutering and awareness of responsible ownership practices.”

- Line 463 onwards: would you not want to link with mental health (next to health)? You can discuss your findings on respondents that were bitten/had family bitten and not liking the presence of stray dogs, which may indicate mental health risks for these respondents?

This is an interesting idea and could make for a useful follow-on study, however any comments made on this topic here would be conjecture as our data does not relate specifically to this. 

- Line 476: yes, and why would you anticipate this lower likeliness? Some brief words on more highly engaged dog owners being more likely to partake in studies on dogs (as discussed well in the limitation section)? Well done that you indicate percentages from other studies for comparison: an addition could be to indicate whether the study approaches differed from yours?

We have amended to now read:

Lines 563 onwards: “Again, these results might be biased, as more highly engaged dog owners may have been more likely to participate in this study and may also be less likely to allow their dogs to roam. The results are higher than those reported in studies using similar sampling approaches (i.e. relying on voluntary participation in questionnaires/interviews) in …”

- Lines 538-545: You end this interesting paragraph with age as a targeting option. Would you not want to add how the attitude aspect that associated neutering in your study could be addressed specifically?

Thank you for this suggestion, we have now included: Lines 638-640 “For example, interventions could target these groups to increase knowledge of the necessity to neuter dogs and possible effects of neutering on dog health and behaviour.”

- Line 548: Mind that questionnaires do not measure on behaviour itself.

Thank you – we’ve taken this out.

Further suggestions

- Abstract (line 20): ‘behaviour and outlook’, but your aim (one sentence lower) regards attitudes of dog owners and non-dog owners & practices of dog owners. Is ‘behaviour and outlook of local communities’ the right step up to your study aim? (Also in line 94 of the Introduction).

Amended

- Abstract (line 23 & 26): This study – this questionnaire: you may want to reword it (esthetical reasons only). 

We have replaced survey for questionnaire throughout for consistency. However we would like to keep questionnaire and study, as the questionnaire (the questions and responses) was part of the study (which also includes description and statistical analyses of responses).

- Abstract (line 35): Can your findings help to ‘inform’ or ‘shape’ management interventions?

Amended

- Introduction (line 61): ‘these factors’: is it clear to the reader which factors?

We have amended to “Organisations involved in dog population management should consider these cultural, religious, and risk factors to ensure interventions are effective” (lines 68-69)

- Introduction (line 83): ‘encouraging’ – but note that legislation may have an effect other than or next to an encouraging effect.

We have amended to “enforcing”.

- If available, indications of ‘other reasons’ (eg Results line 239) would be informative when percentages are high. 

We’ve included some other reasons “such as family illness; a change in circumstances (e.g. birth of new child in home); moving home; owners going on a long trip away; the dog having puppies; or the dog not getting along with other dogs in the household.” (Lines 279-282)

Review 2:

Thank you for the opportunity to review this manuscript that details dog ownership behaviors and attitudes towards free-roaming dogs in Bulgaria, Italy, and Ukraine. The study employed an online questionnaire and recruited participants via social media. The data showed differences in self-reported behaviors and attitudes between countries, and found statistically significant associations between demographic factors such as age, gender, education, and the measured outcomes.

The submitted article satisfies PLOS ONE’s 7 publication criteria. It is very well written and technically sound, and the limitations are adequately addressed in the discussion. There are a few instances in which the authors draw conclusions or make statements that go beyond the reach of their data, but this can be easily fixed with minor editing.

Going beyond the data: At several points in the manuscript the authors discuss the efficacy of free-roaming dog population control (e.g., abstract, lines 37, 95), but efficacy was not assessed in their study. They surveyed attitudes and behaviors, then hypothesized how those behaviors might affect efficacy, but they did not assess it as they claim.

We have amended the text throughout.

Causality is sometimes implied, but causal relationships cannot be determined from the questionnaire data. This is true when the authors describe the relationship between demographic data and attitudes and behavior: I would caution against saying there was a “significant effect of gender/age/education/feeling threatened” on response variables because cause and effect were not assessed. Instead, I suggest saying there was a statistically significant association between gender/age/etc. and the response variable.

We have amended the terminology throughout to avoid any reader confusion.

I have concerns about the advertisements used for recruitment. The four advertisements are very different, and depending on which advertisement a participant saw, their responses may have been biased. For example, Advert 3 includes the language “stray dog overpopulation is a global problem which is of public health, animal welfare and environmental concern,” whereas Advert 1 simply states that they want a wide range of opinions on stray dogs. This is problematic and needs to be addressed in the methods and/or discussion. How were these advertisements distributed? Was Advert 1 used in one country or in one language while Advert 3 was used in another? If so, significant differences cannot necessarily be attributed to region as they could instead be attributed to the condition of the advertisement. If they were randomly distributed across country/language then it may not be a confound, but there could still be an effect of advertisement and this should be discussed.

All adverts were used in all countries using the translated versions of the adverts (i.e. they were randomly distributed across country). We have included in the methods:

“ All adverts were used in all countries to an equal extent, though it is not possible to know which advert respondents had seen.” Lines 134-136.

And

“We used four different adverts with slightly different wording in order to attract as many respondents as possible. The different adverts may have attracted different subsets of people, which could lead to biases in results. Though we were unable to determine which adverts respondents saw, all adverts were distributed equally across the three countries and as such are unlikely to lead to differences between countries” Lines 667-671.

Line 119 – do you mean Power analysis (instead of sample size)?

We prefer to use sample size calculation, as power wasn’t used in the calculation.

Additional requirements:

We have ensured our manuscript meets PLOS ONE’s style requirements.

We have amended our ethics statement: “Prior to completing the questionnaire, all participants were asked to consent (by selecting yes in a tick box) to their responses being collected, stored, and analysed in an anonymised form for the purpose of reports and publication.” Lines 147-148.

3. Thank you for stating the following in the Competing Interests section: "The authors declare that: A.M.M. and S.H. are employed by VIER PFOTEN International, a global animal welfare organisation; L.M.C has received a research grant from VIER PFOTEN International; and L.M.S.’s research has been funded by VIER PFOTEN International."

We confirm that this does not alter our adherence to PLOS ONE policies and have included this in our statement.

---

## [Decision Letter · Decision Letter 1]

14 Feb 2022

PONE-D-21-15770R1Public attitudes towards free-roaming dogs and dog ownership practices in Bulgaria, Italy, and UkrainePLOS ONE

Dear Dr. Smith,

Thank you for submitting your manuscript to PLOS ONE. After careful consideration, we feel that it has merit but does not fully meet PLOS ONE’s publication criteria as it currently stands. Therefore, we invite you to submit a revised version of the manuscript that addresses the points raised during the review process.

I apologise for the late decision regarding your revised submission; there was some delay in getting responses from the reviewers and January was an unusually busy month when I wasn’t able to dedicate as much time to my editorial commitments as I would have liked. I will do my best to speed up the next steps of the procedure.

I recognise that your revision has addressed nearly all reviewer comments. There are a few remaining issues that need addressing, as you will see detailed at the bottom of this message.

We look forward to receiving your revised manuscript.

Kind regards,

I Anna S Olsson, Ph.D.

Academic Editor

PLOS ONE

Journal Requirements:

Additional Editor Comments (if provided):

This paper presents a questionnaire study carried out in three countries: Ukraine, Italy and Bulgaria. However, the author list represents research institutions in Italy and the UK and international organisations based in Austria and Belgium. Please comment on the absence of Bulgarian and Ukrainian researchers as authors of the paper. Also, please justify the choice of the three countries for the questionnaire study.

Statistical analysis: I’m not familiar with the methods you have used for the statistical analysis, but it seems to me that a justification should be presented for the use of different approaches (Bernoulli logistic regression versus ordinal probit models) depending on which outcome variables you analysed.

Results section overall: I appreciate your detailed and comprehensive presentation of the outcome of the statistical analysis, but for the paper to be accessible to the wider readership that the study topic merits, ideally this presentation should be complemented with a more straightforward summary of the results. Please consider summarizing in words the most important associations that you found, potentially as a first paragraph of the Results section. Your paper is likely to be of interest to practitioners in dog population management (e.g. shelter directors, veterinary officials), and it is important that they are able to read and understand the main findings! 

219-313 For readability, consider only including the percentage of responses, and not the n=x of y, in the written description of the results. You are also presenting these results in figures and supplementary material, and in particular the supplementary material is the best option for the additional detail.

Lines 307 Is the last sentence of the figure legend not redundant?

Line 303 Have you used the term CNR earlier and defined it? If it is – as a I think – used here for the first time it should be written out rather than using the abbreviation.

Reviewers' comments:

Reviewer's Responses to Questions

**Comments to the Author**

1. If the authors have adequately addressed your comments raised in a previous round of review and you feel that this manuscript is now acceptable for publication, you may indicate that here to bypass the “Comments to the Author” section, enter your conflict of interest statement in the “Confidential to Editor” section, and submit your "Accept" recommendation.

Reviewer #1: All comments have been addressed

Reviewer #2: (No Response)

2. Is the manuscript technically sound, and do the data support the conclusions?

Reviewer #1: Yes

Reviewer #2: Partly

3. Has the statistical analysis been performed appropriately and rigorously? 

Reviewer #1: Yes

Reviewer #2: Yes

4. Have the authors made all data underlying the findings in their manuscript fully available?

Reviewer #1: Yes

Reviewer #2: Yes

5. Is the manuscript presented in an intelligible fashion and written in standard English?

Reviewer #1: Yes

Reviewer #2: Yes

6. Review Comments to the Author

Reviewer #1: Dear authors,

Thank you for addressing all comments thoroughly and adequately. I have re-read your manuscript with much interest and look forward to seeing it published.

Reviewer #2: (No Response)

7. PLOS authors have the option to publish the peer review history of their article (what does this mean?). If published, this will include your full peer review and any attached files.

Reviewer #1: **Yes: **Ineke R. van Herwijnen

Reviewer #2: No

---

## [Author Response · Author response to Decision Letter 1]

15 Feb 2022

Dear Editor,

We are very grateful to both you and the reviewers for your feedback on the manuscript. We have addressed each of the comments, as outlined below:

Journal Requirements:

We have reviewed the reference list to ensure it is complete and correct. We have added some references with newly included text, outlined below.

Additional Editor Comments (if provided):

This paper presents a questionnaire study carried out in three countries: Ukraine, Italy and Bulgaria. However, the author list represents research institutions in Italy and the UK and international organisations based in Austria and Belgium. Please comment on the absence of Bulgarian and Ukrainian researchers as authors of the paper. Also, please justify the choice of the three countries for the questionnaire study.

VIER PFOTEN International is an international organization with country specific offices within Bulgaria and Ukraine. We consulted with VIER PFOTEN International employees in Bulgaria and Ukraine to understand the challenges in dog population management specific to these countries, and to ensure the questionnaire was appropriate. Whilst we acknowledge these individuals in the acknowledgement section, their contribution did not merit authorship. 

We have now included justification of the choice of the countries in the introduction: “We selected these focal countries due to the networks established with collaborating organisations (VIER PFOTEN International and Istituto Zooprofilattico Sperimentale dell’Abruzzo e del Molise “Giuseppe Caporale”; IZSAM) that provided local knowledge to facilitate data collection. The focal countries are culturally and environmentally distinct, allowing comparison of the collected data between different countries within Europe.”

Statistical analysis: I’m not familiar with the methods you have used for the statistical analysis, but it seems to me that a justification should be presented for the use of different approaches (Bernoulli logistic regression versus ordinal probit models) depending on which outcome variables you analysed.

We have included further detail in the statistical analysis section (i.e. that Bernoulli logistic regression are for binary outcome variables, whereas ordinal regression are used for ordered categorical outcome variables).

Results section overall: I appreciate your detailed and comprehensive presentation of the outcome of the statistical analysis, but for the paper to be accessible to the wider readership that the study topic merits, ideally this presentation should be complemented with a more straightforward summary of the results. Please consider summarizing in words the most important associations that you found, potentially as a first paragraph of the Results section. Your paper is likely to be of interest to practitioners in dog population management (e.g. shelter directors, veterinary officials), and it is important that they are able to read and understand the main findings! 

We have now included a subsection “Summary of statistical associations” which reads:

“Respondents were less likely to answer that they neutered their dog(s) and more likely to answer that they allow their dog(s) to roam if they identified as (i) male, (ii) religious, (iii) owning dogs for practical reasons, (iv) young, and (v) having no schooling or primary education. Respondents were more likely answer that an increase in free-roaming dogs should be prevented if they identified as (i) female, (ii) feeling threatened by free-roaming dogs, (iii) older, and (iv) having more education. Below we report the detailed statistical findings.”

219-313 For readability, consider only including the percentage of responses, and not the n=x of y, in the written description of the results. You are also presenting these results in figures and supplementary material, and in particular the supplementary material is the best option for the additional detail.

We initially had only the percentages in the text when we first submitted the paper, but, as we included the numbers (n=x of y) as requested by reviewer 1 in the first round of feedback from our submission to this journal, we would like to keep this addition to increase the accuracy of information.

Lines 307 Is the last sentence of the figure legend not redundant?

Respondents could choose multiple answers to some of the questions in the questionnaire, whilst in others they could only choose one. We would like to indicate which results were multi answer questions, so would like to keep this sentence for clarity.

Line 303 Have you used the term CNR earlier and defined it? If it is – as a I think – used here for the first time it should be written out rather than using the abbreviation.

Thank you for spotting this – we’ve now included the full term in the introduction when we first mention reproductive control.

The authors have addressed most of my comments, concerns, and questions. However, there are still a few issues with wording that should be corrected for clarity.

Abstract: “Assessing dog population management interventions is important to determine their long-term impact. It is essential to also determine how the attitudes and dog ownership practices within local communities may influence the efficacy of dog population management” (lines 18-21)

The phrasing and positioning of these statements still suggest to me that population management was directly assessed in this study. The statements should either a) be removed entirely or b) be edited and moved to the end of the abstract and discussed as potential future research.

We have now removed these sentences.

Line 95: The red text below needs to be deleted from any methods or sections that talk about what the study did/aimed to do, because it is misleading. You absolutely can talk about future research or how your data could be used to hypothesize on the efficacy of dog population management programs, but that should be in the discussion section.

This study assesses how the attitudes and dog ownership practices within local communities may influence the efficacy of dog population management, by gauging attitudes towards the presence of free-roaming dogs and of dog ownership practices in three European countries – Bulgaria, Italy, and Ukraine. 

We have now removed this and checked throughout to ensure readers are not misled.

---

## [Editor Report · Decision Letter 2]

17 Feb 2022

Public attitudes towards free-roaming dogs and dog ownership practices in Bulgaria, Italy, and Ukraine

PONE-D-21-15770R2

Dear Dr. Smith,

We’re pleased to inform you that your manuscript has been judged scientifically suitable for publication and will be formally accepted for publication once it meets all outstanding technical requirements.

Kind regards,

I Anna S Olsson, Ph.D.

Academic Editor

PLOS ONE
---

## [Editor Report · Acceptance letter]

21 Feb 2022

PONE-D-21-15770R2 

Attitudes towards free-roaming dogs and dog ownership practices in Bulgaria, Italy, and Ukraine 

Dear Dr. Smith:

I'm pleased to inform you that your manuscript has been deemed suitable for publication in PLOS ONE. Congratulations! Your manuscript is now with our production department. 

Kind regards, 

on behalf of

Dr. I Anna S Olsson 

Academic Editor

PLOS ONE